# A systematic review of self-report measures used in epidemiological studies to assess alcohol consumption among older adults

Kjerstin Tevik[1,2]*, Sverre Bergh[2,3], Geir Selbæk[2,4,5], Aud Johannessen[2,6], Anne-S. Helvik[1,2]

1 Department of Public Health and Nursing, Faculty of Medicine and Health Sciences, Norwegian University of Science and Technology (NTNU), Trondheim, Norway, 2 Norwegian National Advisory Unit on Ageing and Health, Vestfold Hospital Trust, Tønsberg, Norway, 3 Research Centre for Age-related Functional Decline and Disease, Innlandet Hospital Trust, Ottestad, Norway, 4 Department of Geriatric Medicine, Oslo University Hospital, Oslo, Norway, 5 Institute of Clinical Medicine, Faculty of Medicine, University of Oslo, Oslo, Norway, 6 Department of Health, Social and Welfare Studies, Faculty of Health and Social Sciences, University of South-Eastern Norway, Vestfold, Norway

* kjerstin.e.tevik@ntnu.no

**Data Availability Statement:** All relevant data are within the manuscript and its Supporting Information files.

## Abstract

### Background

There is a lack of standardization regarding how to assess and categorize alcohol intake in older adults. The aim of this study was to systematically review methods used in epidemiological studies to define drinking patterns and measure alcohol consumption among older adults.

### Methods

A systematic search was conducted in the MEDLINE, PubMed, PsycINFO, EMBASE, and CINAHL databases for studies published from January 2009 to April 2021. Studies were included if they were observational studies with a quantitative design; the mean age of the participants was ≥ 65 years; questionnaires, screening tools, or diagnostic tools were used to define alcohol consumption; and alcohol consumption was self-reported.

### Results

Of 492 studies considered, 105 were included. Among the 105 studies, we detected 19 different drinking patterns, and each drinking pattern had a wide range of definitions. The drinking patterns abstaining from alcohol, current drinking, and risk drinking had seven, 12 and 21 diverse definitions, respectively. The most used questionnaire and screening tools were the quantity-frequency questionnaire, with a recall period of 12 months, and the full and short versions of the Alcohol Use Disorders Identification Test, respectively.

### Conclusion

No consensus was found regarding methods used to assess, define, and measure alcohol consumption in older adults. Identical assessments and definitions must be developed to

**Funding:** The authors received no specific funding for this work.

**Competing interests:** The authors have declared that no competing interests exist.

make valid comparisons of alcohol consumption in older adults. We recommend that alcohol surveys for older adults define the following drinking patterns: lifetime abstainers, former drinkers, current drinkers, risk drinking, and heavy episodic drinking. Standardized and valid definitions of risk drinking, and heavy episodic drinking should be developed. The expanded quantity-frequency questionnaire including three questions focused on drinking frequency, drinking volume, and heavy episodic drinking, with a recall period of 12 months, could be used.

## Introduction

In recent years, epidemiological studies on alcohol consumption in older adults have been carried out quite regularly, especially in the United States of America (USA) [1–5] and Europe [6–9]. Several studies have shown an increase in alcohol consumption and a decrease in the prevalence of abstention among older adults [10–12]. Older adults have also been found to drink more frequently than younger age groups [13, 14]. The reported prevalence of abstaining from alcohol [15, 16], current drinking [17, 18], elevated drinking [19], risk-drinking [20, 21], or heavy drinking [6, 22] among older adults varies within and between countries and between studies. Some of the variation in the findings may be due to the method used when questioning participants about alcohol consumption using different questionnaires. It may also be due to the definition used to categorize different drinking patterns. There is a lack of standardization regarding how to assess and categorize alcohol intake in older adults. The absence of standardized definitions of different drinking patterns makes it difficult to compare findings between studies [23]. In addition, there is no international standard for the number of grams of alcohol in one drink or unit of alcohol [24–26]. The USA uses the term standard drink [24], which is defined as 14 grams of alcohol [24, 27]. The United Kingdom (UK) uses the term unit of alcohol, which is defined as 8 grams of alcohol [24, 27]. This means that a unit of alcohol in the UK is equivalent to 0.564 (just over half) of a standard drink in the USA. In addition, a standard drink or unit of alcohol might be referred to as a beverage [16] or a glass of alcohol [28] in other studies. In this manuscript, we use the term "drink", which corresponds to a standard drink in the USA and a unit of alcohol in the UK/Europe.

Compared with younger adults, older adults are more sensitive to alcohol due to reduced metabolism of alcohol and changed body composition with decreased body water and increased body fat, leading to higher blood alcohol concentration and a prolonged effect of alcohol [29–31]. Thus, lower levels of alcohol may cause more adverse health consequences in older adults than in younger adults [31]. Different levels of alcohol consumption in middle-aged and older adults have shown to increase the risk for death from coronary heart disease (alcohol intake $\geq$ 60 g/day in men and $\geq$ 20 g/day in women) [32], increase the risk of cancer (alcohol intake > 60 g/day in men and > 30 g/day in women) [33], and dementia and Alzheimer's disease (drinking alcohol five or more times in the previous fortnight) [34].

The greater sensitivity to alcohol should affect how risk consumption is defined in older adults, but internationalized threshold values for risk consumption are not defined. However, alcohol consumption guidelines for older adults have been established in recent years in some Western countries [35]. The US guidelines developed by the National Institute on Alcohol Abuse and Alcoholism (NIAAA) [36] recommend that adults over age 65 who are healthy and do not take medications should not drink more than three drinks on a given day or seven drinks in a week. Drinking above these limits for healthy older adults may cause health problems and

be referred to as elevated drinking or risk drinking [36]. However, these recommendations are not internationalized. Because a high proportion of older adults have several chronic health conditions and use medication that may interact negatively in combined use with alcohol, it has been recommended that definitions of risk related to alcohol consumption in older adults include information regarding both current health status and use of medication [23, 37]. Due to the greater sensitivity to health risk of alcohol among older adults, the prevalence of binge drinking in older age is of interest [38, 39]. NIAAA defines binge drinking as consuming five or more drinks among men and four or more drinks among women in about two hours [39]. Assessment of binge drinking is relevant in alcohol surveys of older adults. Furthermore, it may be relevant to distinguish between binging (infrequent heavy) versus spacing (steady daily) drinking patterns [40], and especially among older adults drinking higher weekly volume (i.e., eight drinks or more). These opposite drinking patterns can produce the same weekly alcohol volume [40] but binge drinking may lead to higher risk of negative health consequences than steady daily drinking [41, 42]. In alcohol surveys of older adults, it may also be relevant to ask about the maximum number of drinks consumed in any day, the frequency of subjective drunkenness, drinking context, and duration of drinking occasions [25, 26, 40].

During the last decades there have been several international expert groups and meetings convened to discuss alcohol measurement and drinking patterns in the general adult population [24, 25, 40, 43, 44]. The aim of these expert groups has been to give an overview of the current knowledge on measuring frequency, quantity, and volume of drinking, and make consensus recommendations [24, 25, 40, 43, 44].

In epidemiological studies of alcohol consumption, it is recommended that participants be classified into one of three categories: lifetime abstainer, former drinker, or current drinker [24, 43, 45]. According to the World Health Organization (WHO), a lifetime abstainer can be defined as never having consumed alcohol in their life; a former drinker as not having consumed alcohol in the last 12 months but having consumed alcohol earlier; and a current drinker as drinking alcohol once a year or more [24]. Assessment of alcohol consumption in epidemiological studies can be done through personal face-to-face interviews, telephone interviews, or self-administered questionnaires [24, 43, 44]. The most commonly used methods to define drinking pattern and measure alcohol consumption are, 1) the quantity-frequency (QF) questionnaire, which includes two questions about drinking frequency and the usual number of drinks consumed on drinking days; 2) the graduated quantity-frequency (GQF) questionnaire which includes six questions about frequency of consuming various quantities of drinks; 3) the beverage-specific quantity-frequency (BSQF) questionnaire which includes 18 questions about drinking particular types of alcoholic beverages and the quantity; 4) the last seven days consumption questionnaire, which is a retrospective diary showing how much alcohol a person drank on each of the last seven days; 5) the last occasion questionnaire, which indicates the quantity of alcohol consumed on the last drinking occasion, and 6) the Yesterday method which asks questions about beverage types and sizes of drinks consumed the day before the interview [24, 26, 40, 43, 44, 46, 47].

The QF questionnaire has been widely used to measure alcohol consumption since the early 1950s [44]. The GQF and the BSQF questionnaires measure both volume of alcohol and patterns of drinking, have been used less, but have an advantage over the QF questionnaire which only measure the volume [44]. Previous studies have reported higher estimates of volume and prevalence of high-risk drinking using GQF compared to QF questionnaire and weekly drinking measures [46, 48]. A variation of the QF questionnaire (the 'period-specific normal week' assessment instrument) includes questions about drinking variability and asks about alcohol consumption during a normal week the last year [44]. The alcohol consumption during the week is separated between weekdays and on weekend (i.e., Friday, Saturday, and

Sunday) [44]. This assessment instrument is relevant to use when exploring groups where weekend drinking may vary substantially from drinking during the week [44]. The Yesterday method may have some advantages in groups where daily drinking is common [47]. An Australian study of the general population found the Yesterday method to minimize under-reporting of overall alcohol consumption compared to the QF and GQF questionnaires, and recommended the Yesterday method as a supplement to the QF and GQF questionnaires [47].

When it comes to questions about drinking frequency, it is preferable to ask in terms of pre-specified frequency range categories such as twice a day, daily, 5–6 times a week/nearly every day, 3–4 times a week, 1–2 times a week, 2–3 times a month, once a month, 6–11 times a year, and 1–5 times a year [25]. Furthermore, it is recommended to ask the question in terms of number of drinks per day and not per occasion, since a day may be a more 'objective' measure [25]. Continued drinking past midnight should be defined in the day [25].

According to the WHO and other expert groups, studies of alcohol consumption in general populations should contain items for measuring drinking pattern, volume of consumption, and prevalence and volume of high-risk consumption [24, 25]. The minimum required method is an expanded QF questionnaire that includes three questions asking about 1) abstention (lifetime and past 12 months) and drinking frequency, 2) usual number of drinks on drinking days, and 3) the frequency of heavy episodic drinking occasions in the last year (i.e., consuming five or more drinks [> 60 g alcohol] in a single day) [24, 25]. Including question about heavy episodic drinking can counter underestimates of alcohol consumption from the traditional QF questionnaire [40]. Volume of alcohol consumption and threshold values for risk consumption may be set by using the expanded QF, the GQF, and the BSQF questionnaire [24, 43]. In addition, in alcohol surveys, it is recommended to include some questions on alcohol-related problems, such as the screening tool Alcohol Use Disorders Identification Test (AUDIT) [24, 43]. The AUDIT, with 10 structured questions, aims to identify individuals with hazardous and harmful drinking patterns [49]. The short version of the AUDIT (AUDIT-C) consists of the three first questions of the AUDIT [50].

The length of the period for which the respondents are asked about alcohol consumption is called the recall period. The recall period may vary from the last day to lifetime and influences the responses given and the representativeness of the actual consumption [24, 26, 45]. A recall period of 12 months is recommended when using QF, GQF, or BSQF questionnaires because this recall period provides a more comprehensive picture of alcohol consumption [24, 43]. Especially when linking alcohol consumption with alcohol-related consequences, a recall period of at least 12 months is of importance [25]. Shorter recall period is more prone to miss intermittent heavy drinkers [26]. Seasonal variability will also be minimized with 12 months recall period [26].

Even though there have been several previous efforts regarding the standardization of methods to assess, define, and measure alcohol consumption in the adult general population [24, 25, 40, 43, 44], the standardization has so far almost been absent for the aged population. It is important to increase the attention around the need for standardized methodology in alcohol surveys in the aged population. Thus, the aim of this study is to systematically review methods used in epidemiological studies to define drinking patterns and measure alcohol consumption among older adults.

## Material and methods

The PRISMA 2009 statement was used as a guideline for writing this review [51]. A PRISMA checklist is provided in S1 Checklist. We do not have a published protocol for this systematic review.

## Search strategy and study selection

A librarian conducted a systematic, computerized search in the MEDLINE, PubMed, PsycINFO, EMBASE, and CINAHL databases for articles published from January 2009 to April 2021. The last search was performed April 13, 2021. The following terms were used for searching the databases: 'alcohol drinking'[MeSH Terms] OR ('alcohol'[All Fields] AND 'drinking'[All Fields]) OR 'alcohol drinking'[All Fields]/trends[MeSH Subheading] OR 'alcohol drinking/epidemiology' AND 'Aged: 65+ years'. Articles were exported and managed using EndNote Version 20. In addition, reference lists of included studies were screened to find studies that were not detected in the systematic searches. Studies were included in the review if the following criteria were met:

- Mean age of participants ≥ 65 years

- Observational studies with quantitative design (longitudinal or cross-sectional)

- Questionnaires, screening tools, or diagnostic tools used to define alcohol consumption

- Self-reported use of alcohol consumption

- Published in a scientific referee-based journal and written in English

  Studies were excluded from the review if they were

- Theoretical, qualitative, editorial articles or comments on studies

- Studies conducted in the general population/sample (≥ 18 years, mean age < 65 years) with subgroup analysis of older adults

- Intervention studies

- Review/meta-analysis studies

## Identification of relevant studies

After identification of studies through searching in bibliographic databases and examining reference lists to identify relevant publications not detected through the computerized search, each title and abstract was screened by the first and last author (KT or ASH) to determine potential eligibility. The full-text versions were obtained if it was unclear whether the study met the inclusion criteria. Any uncertainty regarding study eligibility was resolved through discussion between two authors (KT/ASH).

## Data extraction

From the included studies, the first author (KT) extracted information about year of publication; year of data collection; study country; study population/sample; study design; number of participants; age and gender of participants; questionnaires, screening tools, diagnostic tools or guidelines used to define drinking pattern; recall period; definition of drinking pattern; definition of alcohol content (i.e., grams) in one drink; and measure of alcohol consumption. The present review refers to the measure used by the authors in the original articles in the tables.

## Quality assessment

The quality assessment of the included studies was assessed according to nine predefined criteria (see Table 1) [52, 53] by two authors independently (KT and ASH). Disagreement was resolved by discussion between these two authors. A score of 1 was given for +, and a score of

**Table 1. Criteria for assessing quality.**

| Criteria | | Score |
|---|---|---|
| 1 | The aims/objectives of the study clearly described. | +/−/? |
| 2 | Description of inclusion and exclusion criteria and study participant's rate. | +/−/? |
| 3 | Description of study population (age and gender). | +/−/? |
| 4 | Contained information about study setting. | +/−/? |
| 5 | Number of participants > 1000. | +/−/? |
| 6 | Information about non-responders versus responders. | +/−/? |
| 7 | Funding sources or conflicts of interest that may affect the authors' interpretation of the results described or ruled out. | +/−/? |
| 8 | Ethical approval or consent of participants attained. | +/−/? |
| 9 | Longitudinal design. | +/−/? |

+ = score 1; − (minus) = score 0; ? (unclear) = score 0.

0 was given for both −(minus) and ? (? = unclear). The sum score of the quality assessment of each study could vary between 0 and 9.

An overall methodological quality percentage was calculated. Studies who scored $\geq$ 80% of the maximum obtainable points ($\geq$ 8 points) were considered to have strong quality, studies with a score of 70–79% of the maximum obtainable points (7 points) were considered to have good quality, 50–69% fair quality (5 or 6 points) and < 50% poor quality ($\leq$ 4 points) [54].

## Risk of bias in individual studies

We did not assess risk of bias of individual studies as this is a systematic review regarding methods used to define, and measure alcohol consumption and not regarding interventions, prognosis, or etiology.

## Ethics

Ethical approval was not required because the study used secondary data.

# Results

## Literature search and selection

The bibliographic database search identified 2816 articles. After duplicates were removed, 1279 studies were identified. We found 15 additional records in the reference lists of included studies that were not detected through the systematic searches. Each title and abstract of the 1294 studies were screened by two authors (KT or ASH), and the full texts of 492 studies were considered for possible inclusion. Of the 492 full-text studies considered, 105 were included. Fig 1 presents the PRISMA flow diagram [51], which gives an overview of the search strategy and detailed information about studies that were identified, screened, assessed for eligibility, and included in the review.

## Settings and samples

The characteristics of the included studies (N = 105) are presented in one large sample (S1 Table). The sample size of individual studies ranged from 25 to 36,136,889. The mean age of the participants was from 65.0 to 87.4 years, and the age range was 18 to 105 years. Men and women were included in almost all studies, except for six that included only men [15, 55–59] and two that included only women [60, 61].

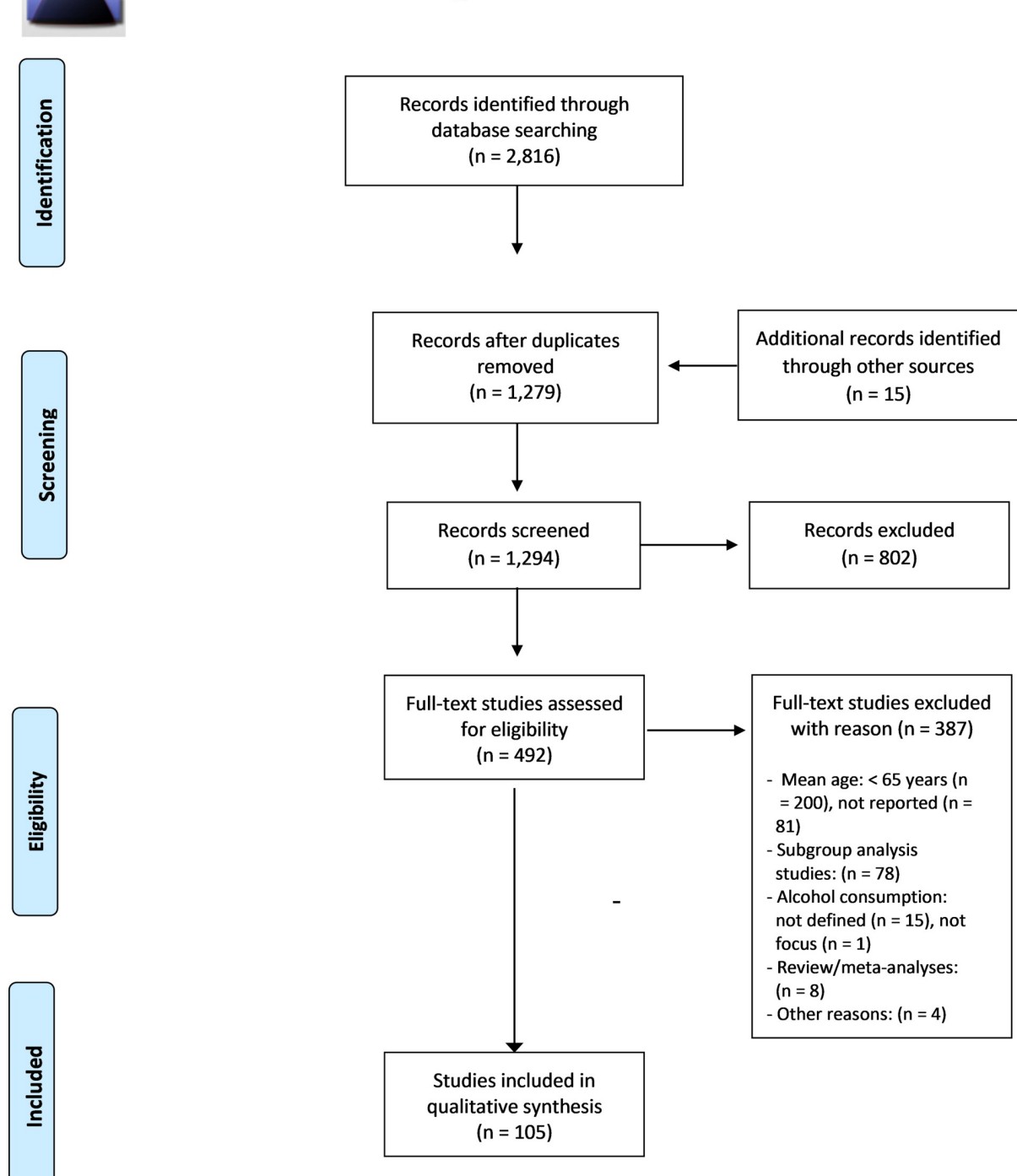

**Fig 1. Flow diagram depicting study identified, screened, assessed for eligibility, and included in this review [51].**

In total, 81 of 105 studies included community-dwelling participants. Eleven studies recruited participants from different health care settings such as hospitals, old-age psychiatry

clinic, an adult psychiatry clinic, and public centers [18, 19, 62–70]. Three studies included both non-institutionalized and institutionalized participants [6, 71, 72], and one study included only Veteran Affairs patients [73]. The study setting and the participants were not clearly described in nine studies [1, 74–81]. In total, 42 of the studies were conducted in Europe, 28 in the USA, 10 in Australia/New Zealand, eight in Latin/South America, 14 in Asia, and one in Africa. Two studies were conducted in both Norway and China.

### Design

Forty-one of the studies had a longitudinal design, and 64 had a cross-sectional design.

### Quality assessment of the included studies

A description of the quality assessment of the included studies is provided in Table 2. Twenty-eight studies received ≥ 8 points indicating strong quality, 38 studies received 7 points (good quality), 36 studies received either 5 or 6 points (fair quality), and three studies received 4 points (poor quality).

### Assessment of alcohol consumption

Alcohol consumption was assessed using the QF questionnaire in 34 studies, the BSQF questionnaire in six studies, and the GQF questionnaire in one study (see Tables 3 and 4). Thirty-six of the studies used either screening tools, diagnostic tools, or guidelines to define different drinking patterns (Tables 3 and 4). In total, eight different screening tools were used, and the AUDIT (4 studies) and short version of the AUDIT (AUDIT-C, 9 studies) were the most used tools. A cutoff value of eight or more was used by three studies applying the AUDIT to define risk drinking [105, 133, 136]. Four of the studies using the AUDIT-C chose a cutoff of four or more in men and three or more in women for hazardous drinking [18, 84, 132] and elevated alcohol consumption [19], respectively (S1 Table). Another study using the AUDIT-C showed a sensitivity and specificity of 94% and 80%, respectively, with a cutoff value of four or more when screening for heavy drinking [50].

The recall period varied from the last week to lifetime (see Table 3). Most of the studies used the last 12 months (N = 34), last month (N = 16), or last week (N = 12) as a recall period. Recall periods of three and six months were used by six studies and one study, respectively. A lifetime recall period was used by four studies. Thirty-five studies did not report the recall period. All studies used self-report to assess alcohol consumption.

A high proportion of the included studies (N = 67) used drinks, units, beverages, or glasses per day, week, or month to measure alcohol consumption (see Tables 3 and 5). In total, 39 of these studies (N = 67) defined the alcohol content in one standard drink, unit, or beverage. The definition of one standard drink, unit, or beverage varied from 8 grams of alcohol to 50 grams of alcohol. The alcohol content in one standard drink or unit was defined as 8 grams, 10 grams, and 12 grams in 6, 16, and 8 studies, respectively. In four studies, a standard drink or unit was defined as 13 and 15 grams of alcohol, respectively, while six studies defined a standard drink or unit as 14 grams of alcohol. Studies conducted in China [109] and Japan [59, 70, 111] defined a standard drink of unit as 50 grams of alcohol [109] and 20–23 grams of alcohol [59, 70, 111], respectively. Three studies used several definitions of one standard drink [59, 99, 118], whereas two studies defined the alcohol content in one drink in ounces [60, 88]. Twenty-eight studies (N = 28) used grams of alcohol per day or week as a measure.

In total, 19 different drinking patterns were detected, which ranged from abstaining to alcohol abuse (see Table 6). Each drinking pattern had diverse definitions. The drinking patterns abstaining from alcohol, current drinking, risk drinking, and heavy drinking had, for example,

**Table 2. Quality assessment of included studies.**

| First author (reference) | 1 Aim | 2 Inclusion/ exclusion | 3 Population (age/gender) | 4 Study setting | 5 n > 1000 | 6 Responders / non-responders comparison | 7 Funding resources/ conflict of interest | 8 Ethical approval/ consent | 9 Longi-tudinal | Score +/-/? Max score 9 |
|---|---|---|---|---|---|---|---|---|---|---|
| Aalto et al. 2011 [50] | + | + | + | + | - | - | + | + | - | 6 |
| Agahi et al. 2016 [6] | + | + | + | + | - | - | + | + | + | 7 |
| Agahi et al. 2019 [74] | + | + | + | ? | + | - | + | + | + | 7 |
| Aguila et al. 2016 [82] | + | + | + | + | + | - | + | + | + | 8 |
| Ahlner et al. 2018 [71] | + | + | + | + | + | - | + | + | - | 7 |
| AlGhatrif et al. 2013 [15] | + | + | + | + | - | + | + | - | + | 7 |
| Almeida et al. 2014 [55] | + | + | + | + | + | - | + | + | + | 8 |
| Almeida et al. 2017 [56] | + | + | + | + | + | - | + | + | + | 8 |
| Barnes et al. 2010 [83] | + | + | + | + | + | - | + | - | - | 6 |
| Bazal et al. 2019 [75] | + | + | + | ? | + | - | + | + | + | 7 |
| Bell et al. 2015 [84] | + | + | + | + | + | - | + | + | + | 8 |
| Britton et al. 2020 [85] | + | + | + | + | + | - | + | + | + | 8 |
| Bryant et al. 2013 [86] | + | + | + | + | + | - | - | - | - | 5 |
| Bryant et al. 2013 [87] | + | + | + | + | + | - | + | - | - | 6 |
| Bryant et al. 2019 [88] | + | + | + | + | + | - | - | + | - | 6 |
| Buja et al. 2010 [77] | + | + | + | ? | + | - | + | - | + | 6 |
| Buja et al. 2011 [76] | + | + | + | ? | + | + | + | + | + | 8 |
| Chan et al. 2010 [62] | + | + | + | + | - | + | + | + | - | 7 |
| Chavez et al. 2016 [73] | + | + | + | + | + | - | + | + | + | 8 |
| Choi et al. 2011 [89] | + | + | + | + | + | - | + | - | - | 6 |
| Cohen-Mansfield et al. 2012 [28] | + | + | + | + | + | - | + | + | - | 7 |
| Cousins et al. 2014 [90] | + | + | + | + | + | - | + | + | - | 7 |
| D'Ovidio et al. 2019 [65] | + | + | + | + | + | - | + | + | - | 7 |
| Davis et al. 2014 [78] | + | + | + | ? | + | + | + | + | - | 7 |
| Dhana et al. 2020 [91] | + | + | + | + | + | - | + | + | + | 8 |
| Forlani et al. 2014 [92] | + | + | + | + | - | - | + | + | - | 6 |
| Foster et al. 2019 [93] | + | + | + | + | + | - | + | + | - | 7 |
| Fuentes et al. 2017 [7] | + | + | + | + | + | - | + | + | - | 7 |
| Gargiulo et al. 2013 [94] | + | + | + | + | + | ? | + | + | + | 8 |
| Gibson et al. 2017 [17] | + | + | + | + | + | + | + | + | - | 8 |
| Gonzalez-Rubio et al. 2016 [95] | + | + | + | + | - | - | + | + | - | 6 |
| Goulden 2016 [1] | + | + | + | ? | + | - | + | + | + | 7 |
| Guidolin et al. 2016 [72] | + | - | + | + | - | - | - | + | - | 4 |
| Hajek et al. 2017 [96] | + | + | + | + | + | - | + | + | - | 7 |
| Halme et al. 2010 [97] | + | + | + | + | + | - | + | - | + | 7 |
| Han et al. 2019 [98] | + | + | ? | + | + | - | + | + | - | 6 |
| Hassing 2018 [8] | + | + | + | + | - | + | + | + | + | 8 |
| Heegard et al. 2011 [16] | + | + | + | + | - | + | - | + | - | 6 |
| Heffernan et al. 2016 [99] | + | + | + | + | - | + | + | + | + | 8 |
| Hoang et al. 2014 [60] | + | + | + | + | + | + | + | + | + | 9 |
| Hoeck et al. 2013 [100] | + | + | + | + | + | - | + | - | - | 6 |
| Holton et al. 2019 [101] | + | + | + | + | + | ? | + | + | + | 8 |
| Hongtong et al. 2016 [102] | + | + | + | + | - | - | + | + | - | 6 |
| Ilomaki et al. 2013 [57] | + | + | + | + | + | - | + | + | - | 7 |
| Ilomaki et al. 2014 [58] | + | + | + | + | + | - | + | + | - | 7 |

*(Continued)*

**Table 2.** (Continued)

| First author (reference) | 1 Aim | 2 Inclusion/ exclusion | 3 Population (age/gender) | 4 Study setting | 5 n > 1000 | 6 Responders / non-responders comparison | 7 Funding resources/ conflict of interest | 8 Ethical approval/ consent | 9 Longi- tudinal | Score +/-/? Max score 9 |
|---|---|---|---|---|---|---|---|---|---|---|
| Immonen et al. 2011 [103] | + | + | + | + | + | + | - | - | - | 6 |
| Immonen et al. 2013 [104] | + | + | + | + | + | - | + | + | - | 7 |
| Iparraguirre 2015 [20] | + | + | + | + | + | - | + | + | + | 8 |
| Ivan et al. 2014 [63] | + | + | + | + | - | - | + | + | - | 6 |
| Jentsch et al. 2017 [61] | + | + | ? | + | + | - | + | + | - | 6 |
| Jeong et al. 2012 [105] | + | - | + | + | - | - | + | + | + | 6 |
| Johannessen et al. 2017 [19] | + | + | + | + | - | + | + | + | - | 7 |
| Kim et al. 2015 [106] | + | + | + | + | + | - | + | + | - | 7 |
| Kim et al. 2020 [66] | + | + | + | + | - | - | + | + | - | 6 |
| Kohno et al. 2019 [81] | + | + | + | - | + | - | + | + | - | 6 |
| Lasebikan et al. 2015 [107] | + | + | + | + | + | - | - | + | - | 6 |
| Li et al. 2017 [108] | + | + | + | + | + | + | + | + | - | 8 |
| Li et al. 2019 [109] | + | + | + | + | + | - | + | + | - | 7 |
| Lima et al. 2009 [22] | + | + | + | + | - | - | + | + | - | 6 |
| Listabarth et al. 2021 [110] | + | + | + | + | + | - | + | - | - | 6 |
| Liu et al. 2019 [111] | + | + | + | + | + | - | + | + | + | 8 |
| Machado et al. 2017 [21] | + | + | + | + | + | - | + | - | - | 6 |
| Marti et al. 2015 [112] | + | ? | ? | + | + | - | + | - | - | 4 |
| McCaul et al. 2010 [113] | + | + | + | + | + | - | + | - | + | 7 |
| McClure et al. 2013 [2] | + | - | + | + | + | - | + | + | - | 6 |
| McEvoy et al. 2013 [3] | + | + | + | + | + | + | + | + | + | 9 |
| Merrick et al. 2011 [114] | + | + | + | + | + | - | - | - | + | 6 |
| Moore et al. 2009 [115] | + | + | + | + | + | - | + | + | - | 7 |
| Munoz et al. 2018 [9] | + | + | + | + | + | ? | + | + | - | 7 |
| Nadkarni et al. 2011 [116] | + | + | + | + | + | - | + | + | - | 7 |
| Nogueira et al. 2013 [117] | + | + | + | + | + | - | + | + | - | 7 |
| Nuevo et al. 2015 [118] | + | + | + | + | + | - | + | + | - | 7 |
| Ormstad et al. 2016 [79] | + | - | + | ? | + | - | + | - | + | 5 |
| Ortola et al. 2017 [119] | + | + | + | + | + | - | + | + | + | 8 |
| Ortola et al. 2019 [120] | + | + | + | + | - | - | + | + | + | 7 |
| Parikh et al. 2015 [121] | + | ? | + | + | + | - | + | - | - | 5 |
| Rao et al. 2015 [122] | + | + | + | + | + | - | + | - | - | 6 |
| Richard et al. 2017 [4] | + | + | + | + | + | - | + | + | + | 8 |
| Roson et al. 2010 [18] | + | + | + | + | + | - | + | + | - | 7 |
| Ryan et al. 2013 [123] | + | + | + | + | + | - | + | - | - | 6 |
| Sacco et al. 2009 [124] | + | + | + | + | + | - | + | - | - | 6 |
| Sanford et al. 2020 [125] | + | + | + | + | + | - | + | + | - | 7 |
| Satre et al. 2011 [64] | + | + | + | + | - | - | + | + | - | 6 |
| Scott et al. 2020 [80] | + | + | + | ? | + | - | + | - | + | 6 |
| Shaw et al. 2011 [126] | + | + | + | + | + | - | + | - | + | 7 |
| Shiotsuki et al. 2019 [67] | + | + | + | + | + | - | + | + | - | 7 |
| Siddiquee et al. 2020 [59] | + | + | + | + | - | - | + | + | - | 6 |
| Soler-Vila et al. 2019 [127] | + | + | + | + | - | - | + | + | + | 8 |
| Suo et al. 2019 [68] | + | + | + | + | + | - | + | + | - | 7 |
| Tait et al. 2013 [128] | + | + | + | + | + | - | + | + | + | 8 |

(*Continued*)

**Table 2.** (Continued)

| First author (reference) | 1 Aim | 2 Inclusion/ exclusion | 3 Population (age/gender) | 4 Study setting | 5 n > 1000 | 6 Responders / non-responders comparison | 7 Funding resources/ conflict of interest | 8 Ethical approval/ consent | 9 Longi-tudinal | Score +/-/? Max score 9 |
|---|---|---|---|---|---|---|---|---|---|---|
| Tateishi et al. 2019 [69] | + | + | + | + | + | - | + | + | + | 8 |
| Tevik et al. 2017 [129] | + | + | + | + | + | - | + | + | - | 7 |
| Tevik et al. 2019 [130] | + | + | + | + | + | + | + | + | + | 9 |
| Towers et al. 2018 [131] | + | + | + | + | + | - | + | - | - | 6 |
| Towers et al. 2019 [132] | + | + | + | + | + | - | + | + | - | 7 |
| Vafeas et al. 2017 [133] | + | + | - | + | - | - | - | + | - | 4 |
| van Oort et al. 2020 [134] | + | + | + | + | - | - | + | + | + | 7 |
| Villalonga-Olives et al. 2020 [135] | + | + | + | + | + | - | ? | + | + | 7 |
| Villar Luis et al. 2018 [136] | + | + | + | + | - | - | - | + | - | 5 |
| Waern et al. 2014 [12] | + | + | + | + | + | + | + | + | - | 8 |
| Wang et al. 2017 [137] | + | + | + | + | + | + | + | + | - | 8 |
| Weyerer et al. 2011 [138] | + | + | + | + | + | - | + | + | + | 8 |
| Weyerer et al. 2009 [139] | + | + | + | + | + | + | + | + | - | 8 |
| Wilson et al. 2014 [5] | + | + | + | + | + | - | + | - | - | 6 |
| Zaitsu et al. 2020 [70] | + | + | + | + | + | ? | + | + | - | 7 |

+ = score 1; – (minus) = score 0; ? (unclear) = score 0.

seven, 12, 21, and 25 diverse definitions, respectively (Table 6). The definitions of abstaining from alcohol ranged from not drinking alcohol at all in their entire life to drinking less than one unit a week. Twenty-three studies separated abstainers from former drinkers when defining abstainers. Current drinkers were defined as drinking alcohol in the last 12 months to consuming ≥ 60 grams of alcohol per day. The definition of risk drinking ranged from drinking eight or more drinks per week for both women and men to drinking 35 and 50 drinks per week for women and men, respectively. Table 6 describes the range of definitions for other drinking patterns. Twenty-one studies defined heavy episodic drinking/binge drinking, and the most used definition was drinking five or more drinks on any occasion within the past 30 days.

## Discussion

This systematic review has reviewed different ways of asking about alcohol consumption in older adults to define and measure alcohol consumption and drinking patterns in epidemio-logical studies. No consensus was found regarding methods used to assess, define, and measure alcohol consumption in older adults. Among the 105 studies included, we detected 19 different drinking patterns, and each drinking pattern had a wide range of definitions. The drinking patterns abstaining from alcohol, current drinking, and risk drinking had seven, 12, and 21 diverse definitions, respectively. The most used questionnaire and screening tools were the QF questionnaire, with a recall period of 12 months, and the AUDIT/AUDIT-C, respectively. The volume of alcohol intake was more frequently presented in standard drinks than in grams, and the definition of one standard drink varied from 8 grams of alcohol to 50 grams of alcohol.

### Definition of drinking patterns

**Abstainers and drinkers.** In alcohol surveys, it is important to ask about drinking fre-quency to identify those who are abstainers and drinkers [43]. This review detected seven and

12 different definitions of abstainers and current drinkers, respectively. The wide variation in definitions will have a significant impact on how these drinking patterns are classified [45]. When the definitions are not identical, we are not able to make a valid comparison between studies of the prevalence of abstainers and current drinkers in older adults.

Twenty-three of the included studies separated abstainers from former drinkers when defining abstainers. It is recommended that individuals in alcohol studies be divided into life-time abstainers, former drinkers, and current drinkers [24, 43]. This is especially important in studies investigating the health consequences of alcohol consumption. If former drinkers are included in the abstaining category, the health benefits of light-to-moderate drinking may be exaggerated [45]. Former drinkers might have quit drinking due to health problems [45, 141], and if they are included in the abstainer category, it may not be the absence of alcohol that ele-vates their risk for negative health consequences, but rather their poor health [141]. The defini-tions provided by the WHO [24] for lifetime abstainers (never having consumed alcohol in their life), former drinkers (not having consumed alcohol in the last 12 months, but having consumed alcohol earlier), and current drinkers (consuming alcohol once a year or more) could be used in alcohol surveys conducted in samples of older adults.

**Risk drinkers.**   This review detected 21 different definitions of risk drinking among older adults. However, a commonly used definition was drinking above seven drinks a week, which is in line with the US alcohol guidelines for older adults developed by the NIAAA [36]. Even so, to our knowledge, this definition has not been validated in a sample of older adults.

The WHO has suggested an international threshold value for high-risk drinking as greater than 60 grams of alcohol (equivalent to 4.3 standard drinks in the USA) on any given day for men and greater than 40 grams of alcohol (equivalent to 2.9 standard drinks in the USA) for women [24]. These values are estimated for the general population, and not for the older popu-lation, who might experience negative health consequences at lower alcohol consumption than younger adults due to alcohol-related physiological changes [31].

An international threshold value for risk drinking has not been set for the older population, but it is highly warranted. A risk-drinking definition for older adults should be developed and validated in epidemiologic observation studies among older adults. In addition, whether a risk definition for older adults should include questions about both health condition and use of medications should be considered. Several authors have recommended that a risk definition for older adults should account for both current health status and use of medication [23, 37, 142].

**Heavy episodic drinking.**   Of the 21 studies including a definition of heavy episodic drink-ing/binge drinking in their assessment of alcohol consumption, the most used definition was consuming five or more drinks of alcohol on any occasion within the past 30 days. It is highly recommended to assess heavy episodic drinking [24], but there is little agreement on how heavy episodic drinking should be defined for older adults. For example, "is drinking on any occasion" the best wording? Others have pointed out that "occasion" is difficult to understand, and the definition could rather use drinks within "one day" to increase precision and compre-hensibility [25, 43]. Furthermore, it is important to develop a standardized definition includ-ing the number of drinks needed to define heavy episodic drinking for older adults, especially because tolerance is reduced in older adults [29–31].

## Assessment of alcohol consumption

**Questionnaires.**   Most (N = 34) of the studies used the QF questionnaire to assess alcohol consumption. This questionnaire is commonly used in assessment of alcohol consumption [44, 143], but has been criticized for underestimating alcohol consumption compared with the GQF questionnaire [46, 143, 144]. Assessing alcohol consumption only with the use of the QF

questionnaire, could lead to heavy episodic drinkers not being identified [46, 145]. A previous study by Rehm et al. [46] has also shown that the GQF questionnaire was superior to QF questionnaire and weekly diary in capturing risky and harmful drinking volumes. Thus, the WHO recommends that with the use of the QF questionnaire, a question about heavy episodic drinking should be included when estimating the volume of alcohol consumption and the proportion of risk drinking. As already mentioned in the introduction, this questionnaire is called the expanded QF questionnaire [24]. Only two studies in this review followed the WHO's recommendation when using the QF questionnaire [57, 58]; thus, there is room for improvement in studies using the QF questionnaire to assess alcohol consumption in older adults.

In the present study, we found that few studies used the GQF (N = 1) and BSQF (N = 6) questionnaires. With the GQF questionnaire, the proportion of older adults engaging in risk consumption can be estimated without any additional questions [24], and the GQF is recommended by the WHO and other expert groups for use in the general population [24, 40]. As the GQF (six questions) and the BSQF (18 questions) questionnaires include more questions than the expanded QF (three questions) questionnaire, it will be more time consuming for older adults to respond to all questions in the GQF and the BSQF questionnaires. In addition, older adults might find it difficult to answer the questions included in the GQF and BSQF, such as frequencies of consuming various quantities of drinks and drinking particular types of beverages. Thus, the response rate might be lower with the use of the GQF and BSQF questionnaires than with the expanded QF questionnaire. The results will be systematically biased if older adults do not respond to difficult questions regarding alcohol intake, or if they answer them in a systematically biased way [146]. A review has examined how aging affects self-report questionnaires in general [146], and the authors found that the completeness of self-report questionnaires among older adults decreased with increasing difficulty of questions. To ensure higher response rate and accuracy in alcohol surveys in older adults, the best approach might be to use the expanded QF questionnaire.

Moreover, underestimation of alcohol consumption, especially among heavy drinkers, is well known [147] and will influence the validity of the results. In addition, older adults who do not drink alcohol at all or who do so at very low levels, might not answer the questions about alcohol consumption, as they might consider the questions to be irrelevant. Older adults might not answer alcohol questions due to stigma associated with drinking [148, 149], and answers to alcohol questions by older adults are prone to recall bias due to cognitive impairment and memory errors [150–152]. Thus, the validity of the results might increase with a face-to-face interview when using the QF, GQF, or BSQF questionnaire, where the interviewer can help the participants to complete the questions and to recall the number of alcohol drinks consumed. The use of pictures of standard drinks might be valuable in this context [24, 26]. A face-to-face interview is also recommended in the general population [43]. However, in large population-based studies, face-to-face interviews might not be feasible as they are more time consuming and costly than self-administered questionnaires [153].

**Screening tools.** In total, eight different screening tools were used to assess alcohol consumption, the most common of which were the AUDIT and AUDIT-C. A cutoff value of eight or more was used by several studies included in this review [105, 133, 136]. A meta-analysis [154] has also shown that there is strong evidence for the diagnostic accuracy of the AUDIT with a cutoff value of eight or more in elderly patients [154]. However, the WHO [49] recommends a cutoff value of seven or more for women and men 65 years or older which will increase the sensitivity for this population [49]. The short version of the AUDIT (AUDIT-C) might have an optimal cutoff value of four or more, as one of the included studies showed high sensitivity and specificity with the use of this cutoff value when screening for heavy drinking [50].

In alcohol surveys including older adults, the AUDIT or AUDIT-C work well and are recommended [49, 50]. However, the screening tool CARET (Comorbidity Alcohol Risk

**Table 3. Method and measure of alcohol consumption in older adults.**

| Author and country | Recall period | Data collection | Questionnaire/screening tool/diagnostic tool/ guidelines | Measure |
|---|---|---|---|---|
| Aalto et al. 2011 Finland [50] | Last month | Self-reporting | AUDIT | g/week, drinks/week score of screening |
| Agahi et al. 2016 Sweden [6] | Last 12 months | Face-to-face interview | QF | drinks/month |
| Agahi et al. 2019 Sweden [74] | Last 12 months | Face-to-face interview | Drinking frequency | times/week |
| Aguila et al. 2016 Mexico [82] | Last month | Face-to-face interview | QF, NIAAA guidelines | drinks/week |
| Ahlner et al. 2018 Sweden [71] | Last month | Self-reporting | Drinking frequency Weekly volume of alcohol: beer, wine, spirits | days/week g/week |
| AlGhatrif et al. 2013 USA [15] | Last month | In-home interviews | QF | drinks/month |
| Almeida et al. 2014 Australia [55] | Last week | Face-to-face interview | Weekly quantity of drinks | drinks/week |
| Almeida et al. 2017 Australia [56] | Last week | Face-to-face interview | Weekly quantity of drinks | drinks/week |
| Barnes et al. 2010 USA [83] | Last 12 months | Mailed survey | CARET | score of screening |
| Bazal et al. 2019 Spain [75] | Not reported | Self-reporting | Validated food-frequency questionnaire | g/day g/week units/week |
| Bell et al. 2015 England [84] | Last 12 months | Self-reporting | Drinking frequency Daily quantity of drinks AUDIT | times/week drinks/day score of screening |
| Britton et al. 2020 United Kingdom [85] | Last week AUDT-C: not reported | Self-reporting | Weekly volume and weekly quantity of units of alcohol AUDIT-C | g/week, units/week score of screening |
| Bryant et al. 2013 USA [86] | Last 12 months | Telephone survey | QF Frequency of binge drinking | drinks/day days/year |
| Bryant et al. 2013 USA [87] | Last 12 months | Telephone survey | Frequency of binge drinking | drinks/sitting, days/year |
| Bryant et al. 2019 USA [88] | Last month | Telephone interview | QF | drinks/month |
| Buja et al. 2010 Italy [77] | Last week | Face-to-face interview | Daily volume of wine or beer Weekly volume of spirits | g/day g/week |
| Buja et al. 2011 Italy [76] | Last 12 months | Face-to-face interview | Daily volume of wine or beer Weekly volume of spirits | g/day g/week |
| Chan et al. 2010 China [62] | Not reported | Face-to-face interview | BSQF | g/week |
| Chavez et al. 2016 USA [73] | Last 12 months | Face-to-face interview | AUDIT-C | score of screening |
| Choi et al. 2011 USA [140] | Last 3 months | Self-reporting | QF and binge drinking | drinks/day |
| Cohen-Mansfield et al. 2012 Israel [28] | Last month | In-home interview | QF | glass of different beverages/day |
| Cousins et al. 2014 Ireland [90] | Last 6 months | Self-reporting | QF, NIAAA guidelines CAGE | drinks/day, drinks/week, score of screening |
| D'Ovidio et al. 2019 Ireland, the Netherlands, and Italy [65] | Lifetime | Self-reporting | Weekly quantity of units/glasses of alcohol | Lifetime total units of alcohol |
| Davis et al. 2014 Iceland [78] | Last month | Self-reporting | QF | g/week, drinks/week |
| Dhana et al. 2020 USA [91] | Not reported | Self-reporting | Food frequency questionnaire | drinks/day, g/day |
| Forlani et al. 2014 Italy [92] | Not reported | Face-to-face interview | Daily quantity of each beverage | units/day |
| Foster et al. 2019 England [93] | Last week | Self-reporting | Drinking diary: weekly quantity of units of alcohol | units/week |
| Fuentes et al. 2017 Europe [7] | Last 3 months | Self-reporting | QF and binge drinking | drinks/occasion |
| Gargiulo et al. 2013 Italy [94] | Not reported | In-home-interview | Daily volume of wine drinking | ml/day |
| Gibson et al. 2017 Jamaica [17] | Last 12 months | Self-reporting | Assessed drinking alcohol last month | not reported |
| Gonzalez-Rubio et al. 2016 Spain [95] | Last 12 months | Interviewed by a trained nutritionist | QF | g/day |
| Goulden 2016 USA [1] | Last 3 months | Telephone/In person interview | QF | g/day |

*(Continued)*

**Table 3.** (Continued)

| Author and country | Recall period | Data collection | Questionnaire/screening tool/diagnostic tool/guidelines | Measure |
|---|---|---|---|---|
| Guidolin et al. 2016 Brazil [72] | Last 12 months | In-home/In-hospital interview | Mini International Neuropsychiatric Interview | diagnostic criteria |
| Hajek et al. 2017 Germany [96] | Not reported | Computer assisted interview | Drinking frequency | times/week |
| Halme et al. 2010 Finland [97] | Last 12 months | Self-reporting | BSQF | drinks/week |
| Han et al. 2019 USA [98] | Last month | Self-reporting | Binge drinking | drinks/occasion |
| Hassing 2018 Sweden [8] | Not reported | Self-reporting | BSQF | drinks/week |
| Heegaard et al. 2011 Denmark [16] | Last week | Face-to-face interview | Weekly quantity of different beverages | beverages/week |
| Heffernan et al. 2016 Australia [99] | Last 12 months | Face-to-face interview | QF | drinks/day |
| Hoang et al. 2014 USA [60] | Last month | Self-reporting | QF | drinks/week |
| Hoeck et al. 2013 Belgium [100] | Last week | Self-reporting | Weekly quantity of glasses of alcohol CAGE | glasses/week score of screening |
| Holton et al. 2019 Ireland [101] | Not reported | Self-reporting | QF | drinks/week |
| Hongthong et al. 2016 Thailand [102] | Not reported | Face-to-face interview | Quantity per drinking episode Drinking frequency | drinks/drinking episode days/week |
| Ilomaki et al. 2013 Australia [57] | Last 12 months | Self-reporting | Expanded QF (including binge drinking) CAGE | drinks/day score of screening |
| Ilomaki et al. 2014 Australia [58] | Last 12 months | Self-reporting | Expanded QF (including binge drinking) CAGE | drinks/day score of screening |
| Immonen et al. 2011 Finland [103] | Not reported | Self-reporting | QF | drinks/day, drinks/week |
| Immonen et al. 2013 Finland [104] | Not reported | Self-reporting | QF | drinks/day, drinks/week |
| Iparraguirre et al. 2015 England [20] | Not reported | Self-reported | Guidelines: National Institute for Health and Care Excellence | units/week |
| Ivan et al. 2014 USA [63] | Last month | Self-reporting | Weekly quantity of drinks | drinks/week |
| Jentsch et al. 2017 Germany [61] | Not reported | Telephone survey/self-report | AUDIT-C | score of screening |
| Jeong et al. 2012 Korea [105] | Last 12 months | Clinical interview | AUDIT DSM-IV | drinks/week, score of screening diagnostic criteria |
| Johannessen et al. 2017 Norway [19] | Not reported | Face-to-face interview | AUDIT, AUDIT-C | score of screening |
| Kim et al. 2015 Korea [106] | Not reported | Face-to-face interview | AUDIT | score of screening |
| Kim et al. 2020 Korea [66] | Last 12 months and lifetime | Face-to-face interview | QF | standard drinks/week g/week |
| Kohno et al. 2019 Japan [81] | Not reported | Face-to-face interview | Weekly drinking frequency. Assessed type of alcohol and daily quantity of grams of alcohol according to a formula | days/week g/day |
| Lasebikan et al. 2015 Nigeria [107] | Last week | Face-to-face interview | Daily and weekly quantity of units of alcohol | units/day, units/week |
| Li et al. 2017 China and Norway [108] | Last 12 months | Self-reporting | Drinking frequency (Norway) | time/month |
| Li et al. 2019 China and Norway [109] | China: Not reported Norway: Last 12 months | Self-reporting | China: Daily and weekly quantity of drinks (liquor, wine, beer) Norway: Drinking frequency | China: drinks/day, drinks/week Norway: days/week |
| Lima et al. 2009 Brazil [22] | Not reported | Self-reporting | Daily quantity of beverages | drinks/day |
| Listabarth et al. 2020 12 European countries [110] | Last week Last 3 months | Computer assisted personal interview | Weekly quantity of drinks Frequency of excessive drinking | drinks/occasion |
| Liu et al. 2019 Japan [111] | Not reported | Self-reporting | QF | units/day |

(*Continued*)

**Table 3.** (Continued)

| Author and country | Recall period | Data collection | Questionnaire/screening tool/diagnostic tool/ guidelines | Measure |
|---|---|---|---|---|
| Villar Luis et al. 2018 Brazil [136] | Not reported | In home interview | AUDIT, MAST-G | score of screening |
| Machado et al. 2017 Latin America [21] | Not reported | Self-reporting | QF, NIAAA guidelines | units/week |
| Marti et al. 2015 USA [112] | Last month | Computer assisted interview | Binge drinking | drinks/occasion |
| McCaul et al. 2010 Australia [113] | Not reported | Self-reporting | Drinking frequency Quantity of drinks per day | days/week drinks/day |
| McClure et al. 2013 USA [2] | Last 12 months | Self-reporting | QF Binge drinking | drinks/week drinks at one sitting |
| McEvoy et al. 2013 USA [3] | Not reported | Self-reporting | Weekly quantity of different beverages NIAAA guidelines | g/week, drinks/day |
| Merrick et al. 2011 USA [114] | Last 12 months | Computer assisted interview | QF, heavy episodic drinking, NIAAA guidelines | drinks/month, drinks/day |
| Moore et al. 2009 USA [115] | Last 12 months | Face-to-face interview | QF | drinks/week |
| Munoz et al. 2018 Europe [9] | Last 12 months | Computer assisted interview | Composite International Interviews, DSM-IV | diagnostic criteria |
| Nadkarni et al. 2011 Dominican Republic [116] | Not reported | Face-to-face interview | Weekly quantity of units of alcohol | units/week |
| Nogueira et al. 2013 Brazil [117] | Not reported | Face-to-face interview | Screening for lifetime alcohol abuse and dependence (SRQ) | score of screening |
| Nuevo et al. 2015 Europe [118] | Last week | Face-to-face interview | Timeline-follow-back method | standard drinks/week |
| Ormstad et al. 2016 Norway [79] | Not reported | Self-reporting | Drinking frequency | times/week |
| Ortola et al. 2017 Spain [119] | Last 12 months | In-home interview | Validated diet history Binge drinking | g/day g/drinking occasion |
| Ortolà et al. 2019 Spain [120] | Last 12 months | Computer assisted telephone interview | Validated diet history (34 alcoholic beverages) | g/day |
| Parikh et al. 2015 USA [121] | Last month | Self-reporting | Binge drinking | drinks/occasion |
| Rao et al. 2015 United Kingdom [122] | Not reported | Self-reporting | Weekly quantity of units of alcohol | units/week |
| Richard et al. 2017 USA [4] | Last 12 months | Self-reporting | QF | times/week, drinks/day |
| Roson et al. 2010 Spain [18] | Not reported | Self-reporting | QF AUDIT, AUDIT-C, SIAC | drinks/week, g/week score of screening |
| Ryan et al. 2013 USA [123] | Last 12 months | Face-to-face interview | QF, heavy episodic drinking, NIAAA guidelines | drinks/month, drinks/day |
| Sacco et al. 2009 USA [124] | Last 12 months | Face-to-face interview | QF, NIAAA guidelines DSM-IV | drinks/week, drinks/ occasion diagnostic criteria |
| Sanford et al. 2020 USA [125] | Last 12 months | Self-reporting | Daily quantity of drinks Binge drinking | drinks/day |
| Satre et al. 2011 USA [64] | Last 30 days | Telephone interview | Drinking frequency GQF SMAST Heavy episodic drinking | days/month drinks/month score of screening drinks/occasion |
| Scott et al. 2020 USA [80] | Last 3 months | Self-reporting | QF | drinks/day, drinks/week |
| Shaw et al. 2011 USA [126] | Last month | Self-reporting | QF, NIAAA guidelines | drinks/month |
| Shiotsuki et al. 2019 Japan [67] | Not reported | Self-reporting | Daily volume of alcohol prior to stroke onset | g/day |
| Siddiquee et al. 2020 Japan [59] | Last week/ month | Self-reporting | BSQF | g/day |
| Soler-Vila et al. 2019 Spain [127] | Last year Last month | Computer assisted telephone interview | Validated diet history: daily quantity of drinks and volume of alcohol and binge drinking | drinks/day g/day g/session |

*(Continued)*

**Table 3.** (Continued)

| Author and country | Recall period | Data collection | Questionnaire/screening tool/diagnostic tool/ guidelines | Measure |
|---|---|---|---|---|
| Suo et al. 2019 China [68] | Lifetime | Electronic questionnaire | Yearly volume of alcohol (spirits, beer, wine) (cumulative drinking amount) | g/day-years |
| Tait et al. 2013 Australia [128] | Not reported | Self-reporting | QF, Australian guidelines | drinks/day |
| Tateishi et al. 2019 Japan [69] | Not reported | Self-reporting | Daily volume of alcohol consumption | g/day |
| Tevik et al. 2017 Norway [129] | Last 12 months | Self-reporting | Drinking frequency | days/week |
| Tevik et al. 2019 Norway [130] | Last 12 months | Self-reporting | Drinking frequency | days/week |
| Towers et al. 2018 New Zealand [131] | Not reported | Self-reporting | AUDIT-C | drinks/day |
| Towers et al. 2019 New Zealand [132] | AUDIT-C: 12 months CARET: Not reported | Self-reporting | AUDIT-C, CARET | AUDIT-C: score of screening CARET: not reported cut-off value |
| Vafeas et al. 2017 Australia [133] | Not reported | Self-reporting | AUDIT | score of screening |
| van Oort et al. 2020 The Netherlands [134] | Last 12 months | Self-reporting | Validated food-frequency questionnaire | g/week |
| Villalonga-Olives et al. 2020 USA [135] | Last 3 months | Self-reporting | Frequency of binge drinking | drinks/occasion |
| Waern et al. 2014 Sweden [12] | Last week | Psychiatric interview | Drinking frequency Weekly volume of alcohol intake | times/week g/week |
| Wang et al. 2017 China [137] | Last 12 months | Face-to face interview | Drinking frequency | days/week |
| Weyerer et al. 2009 Germany [139] | Not reported | Structured clinical interview | BSQF, Guidelines: British Medical Association | g/day |
| Weyerer et al. 2011 Germany [138] | Not reported | Structured clinical interview | BSQF | g/day |
| Wilson et al. 2014 USA [5] | Not reported | Self-reporting | QF, NIAAA guidelines, binge drinking ARPS Risk Classification Algorithm | times/week drinks/week, drinks/day score of screening |
| Zaitsu et al. 2020 Japan [70] | Lifetime | Self-reporting | Daily number of drinks and drinking years | drinking years |
| | | | | drinks–years |
| | | | | drinks/day |

ARPS = Alcohol Related Problem Survey; AUDIT = Alcohol Use Disorders Identification Test; BSQF = Beverage Specific Quantity-Frequency; CAGE = Cut down, Annoyed, Guilty, Eye opener; CARET = Comorbidity Alcohol Risk Evaluation Tool; DSM-IV = Diagnostic and Statistical Manual of Mental Disorders; GQF = Graduated Quantity-Frequency; ICD = International Classification of Diseases; MAST-G = Michigan Alcoholism Screening Test–Geriatric Version; NIAAA = National Institute on Alcohol Abuse and Alcoholism; QF = Quantity Frequency; SIAC = Systematic Inventory Alcohol consumption questionnaire; SMAST = Short Michigan Alcohol Screening Test.

Evaluation), which includes both health condition and use of medication when assessing alcohol risk, could also be relevant to use in an elderly sample [83, 132].

## Guidelines

Several of the included studies (N = 11) used alcohol guidelines to assess and define the drinking pattern of older adults. However, alcohol guidelines are not internationally standardized [35]. Some of the studies used guidelines for the general population [20], whereas other used guidelines for older adults [126]. Because a drink of alcohol is not standardized, it might also be difficult to standardize alcohol guidelines for older adults.

**Table 4. Methods used to define drinking pattern.**

| Method | N studies* |
|---|---|
| QF | 34 |
| BSQF | 6 |
| GQF | 1 |
| Daily quantity (units) or volume (gram) of alcohol | 13 |
| Weekly quantity (drinks/glasses/units) or volume (gram) of alcohol | 17 |
| Yearly alcohol consumption (g/day-years, drink/years) | 2 |
| Drinking frequency | 16 |
| Binge drinking/heavy episodic drinking | 18 |
| Screening tools** | 21 |
| Diagnostic tools*** | 4 |
| Guidelines | 11 |
| Food-frequency questionnaire | 3 |
| Drinking diary | 1 |
| Diet history | 2 |
| Other | 4 |

QF = Quantity-Frequency; BSQF = Beverage Specific Quantity-Frequency; GQF = Graduated Quantity-Frequency.

*The number of studies does not sum up to 105 as several studies used several methods to define drinking pattern.

**ARPS (Alcohol Related Problem Survey) Risk Classification Algorithm; AUDIT (Alcohol Use Disorders Identification Test); AUDIT-C (short version); CAGE (Cut down, Annoyed, Guilty, Eye opener); CARET (Comorbidity Alcohol Risk Evaluation Tool); MAST-G (Michigan Alcoholism Screening Test-Geriatric version); SMAST (Short Michigan Alcohol Screening Test); SRQ (Screening for lifetime alcohol abuse and dependence).

***DSM-IV (Diagnostic and Statistical Manual of Mental Disorders); Mini International Neuropsychiatric Interview.

**Table 5. Different measures of alcohol consumption.**

| Measure | N studies* |
|---|---|
| drinks/standard drinks/units/beverages per day | 26 |
| drinks/standard drinks/units/beverages/glasses per week | 34 |
| drinks/month | 7 |
| drinks on one occasion/sitting/drinking episode | 11 |
| g/day | 17 |
| g/week | 11 |
| g/drinking occasion/session | 2 |
| ml/day | 1 |
| times/week | 7 |
| days/week | 9 |
| days or time/month | 2 |
| days/year | 2 |
| drink-years | 1 |
| g/day–years | 1 |
| lifetime total units of alcoholic beverages | 1 |
| score of screening or diagnostic criteria | 23 |

g = gram; ml = milliliters.

*The number of studies does not sum up to 105 as several studies used several measures of alcohol consumption.

**Table 6. Definitions of different pattern of alcohol consumption.**

| Category | Definition | Author and country |
|---|---|---|
| **ABSTAINERS** | | |
| Lifetime abstainers | Not drinking alcohol in their entire life | AlGhatrif et al. 2013 USA [15] |
| Lifetime abstainers | Never consumed alcohol | Jentsch et al. 2017 Germany [61] Zaitsu et al. 2020 Japan [70] |
| Lifetime abstainers | < 12 drinks in life | Sanford et al. 2020 USA [125] |
| Lifetime abstainers | Not defined | Towers et al. 2019 New Zealand [132] |
| Abstainer | Denied drinking alcohol | Scott et al. 2020 USA [80] |
| Abstainers | Did not drink alcohol | Agahi et al. 2016 Sweden [6] |
| Abstainers | No current or past alcohol consumption | Buja et al. 2011 Italy [76] |
| Abstainers | Not drinking at all during the last year | Halme et al. 2010 Finland [97] |
| Abstainers | No alcohol use in past 12 months | Heffernan et al. 2016 Australia [99] |
| Abstainers | Did not drink at all during the previous year | Lima et al. 2009 Brazil [22] |
| Abstainers | < 1 beverage/week | Heegaard et al. 2011 Denmark [16] |
| Never drinkers | Never consumed/drunk alcohol | Tevik et al. 2019 Norway [130], Siddiquee et al. 2020 Japan [59] |
| Never drinkers | Average alcohol intake of 0 g/day | Ortolà et al. 2019 Spain [120] |
| Never drinkers | Had not consumed > 12 alcoholic drinks during their lifetime | Ilomaki et al. 2013 Australia [57], Ilomaki et al. 2014 Australia [58] |
| Never drinker | Not defined | Zaitsu et al. 2020 Japan [70] |
| Never | Not defined | D'Ovidio et al. 2019 Ireland, the Netherlands, and Italy [65] |
| Current abstainers | Not defined | Towers et al. 2019 New Zealand [132] |
| **NON-DRINKERS** | | |
| Non-drinkers | Life-time abstainers and those who did not drink within the last year (former drinkers) | Richard et al. 2017 USA [4] |
| Non-drinkers | Being abstinent from alcohol a period of 4 years | Goulden et al. 2016 USA [1] |
| Non-drinkers last year | Not consumed alcohol last year | Tevik et al. 2019 Norway [130] |
| Non-drinkers or occasional drinkers | Consumed a mean of 0 glasses alcohol/week | Hoeck et al. 2013 Belgium [100] |
| Non-drinkers | 0 g alcohol/week | Chan et al. 2010 China [62] van Oort et al. 2020 The Netherlands [134] |
| Non-drinkers | 0 drinks/week | Hoang et al. 2014 USA [60] |
| Non-drinkers | Life-long abstainers and very occasional drinkers (individuals who reported 0 g/day of alcohol intake in the last year, but self-described as drinkers) | Soler-Vila et al. 2019 Spain [127] |
| Non-drinker | Did not drink alcohol last year | Britton et al. 2020 United Kingdom [85] |
| Non drinker | Not defined | Kim et al. 2020 Korea [66] |
| Non-drinkers | Not defined | Liu et al. 2019 Japan [111] |
| Non-drinker | Not defined | Shiotsuki et al. 2019 Japan [67] |
| **FORMER DRINKERS** | | |
| Former drinkers | Consumed alcohol in the past, but did not consume any alcohol during the previous 12 months | Ilomaki et al. 2013 Australia [57], Ilomaki et al. 2014 Australia [58] |
| Former drinkers | Drank alcohol during their entire life, but not the past month | AlGhatrif et al. 2013 USA [15] |
| Former drinkers | Consumed alcohol in the past, but no longer at the time of the interview | Buja et al. 2011 Italy [76] |
| Ex-drinkers | Drank previously, but not in the past 12 months | Marti et al. 2015 USA [112] |
| Ex-drinkers | Drank in the past, but stopped drinking currently | Chan et al. 2010 China [62] |
| Former drinkers | Previously reported consumption, but none in the most recent phase | Britton et al. 2020 United Kingdom [85] |
| Former drinker | Used to drink regularly, but have not drunk in the past year | Kim et al. 2020 Korea [66] |
| Former drinkers | No drinks in the past year | Sanford et al. 2020 USA [125] |
| Former drinker | Not defined | Zaitsu et al. 2020 Japan [70] |

*(Continued)*

**Table 6.** (Continued)

| Category | Definition | Author and country |
|---|---|---|
| Former drinker | Not defined | D'Ovidio et al. 2019 Ireland, the Netherlands, and Italy [65] |
| Ex drinkers | Quit before interview | Siddiquee et al. 2020 Japan [59] |
| Ex-drinkers | Average alcohol intake of 0 g/day who answered that they used to drink but had quit | Ortolà et al. 2019 Spain [120] |
| Ex drinkers | Quit drinking and reported 0 g/day of alcohol intake last 12 months | Soler-Vila et al. 2019 Spain [127] |
| Ex-drinkers | Stopped drinking alcohol for at least two years before the interview date | Suo et al. 2019 China [68] |
| **CURRENT DRINKERS** | | |
| Current alcohol use | Alcohol consumption last 12 months | Gibson et al. 2017 Jamaica [17] |
| Current drinking | Alcohol consumption last 12 months | Wang et al. 2017 China [137] |
| Current drinkers | $\geq$ 1 drink in the past year | Sanford et al. 2020 USA [125] |
| Current drinkers | Alcohol consumption last 6 months | Cousins et al. 2014 Ireland [90] |
| Current drinkers | Consumed alcohol at the time of the interview | Buja et al. 2011 Italy [76] |
| Current drinkers | At least alcohol consumption a few times a year | Tevik et al. 2019 Norway [130] |
| Current drinkers | $\geq$ 12 drinks during the previous 12 months | Ilomaki et al. 2014 Australia [58] |
| Current drinkers | Drunk alcohol in the past week or month | Siddiquee et al. 2020 Japan [59] |
| Current drinkers | > 1 drink/week | Roson et al. 2010 Spain [18] |
| Current drinkers | Average alcohol intake > 0 g/day | Ortolà et al. 2019 Spain [120] |
| Current drinker | Definition 1: 40–59 g/day<br>Definition 2: $\geq$ 60 g/day | Shiotsuki et al. 2019 Japan [67] |
| Current drinker | Not defined | Zaitsu et al. 2020 Japan [70] |
| Current drinker | Not defined | D'Ovidio et al. 2019 Ireland, the Netherlands, and Italy [65] |
| Present alcohol consumption | China: Drank alcohol at present; Norway: Alcohol consumption $\geq$ once a month | Li et al. 2017 China and Norway [108] |
| Past year alcohol consumption | Consumed any alcohol in the past 12 months | Bryant et al. 2013 USA [86] |
| Minimal/non-users | < 1 drink/month | Immonen et al. 2013 Finland [104] |
| Lifetime drinkers | Drinking any type of alcoholic beverages $\geq$ 12 times during their lifetime | Munoz et al. 2018 Europe [9] |
| **OCCASIONAL DRINKERS** | | |
| Occasional drinkers | < 4 drinks/month | Gonzàles-Rubio et al. 2016 Spain [95] |
| Occasional drinkers | Drinking at least once over a period of 4 years, but less than once a week | Goulden 2016 USA [1] |
| Occasional drinkers | < 1 time a month, 1–3 times a month, once a week or several times a week | Hajek et al. 2017 Germany [96] |
| Occasional drinking | < 1 drink/week | Hassing 2018 Sweden [8] |
| Occasional drinkers | At least 1 standard drink but < 15 (< 12 for women) in a single week or < 5 (< 4 for women) on the same day | Nuevo et al. 2015 Europe [118] |
| Occasional drinkers | Including nondrinkers and < 1 day/week | Kohno et al. 2019 Japan [81] |
| Occasional drinkers | Alcohol consumption a few times a year | Tevik et al. 2019 Norway [130] |
| Occasional drinker | Not defined | Shiotsuki et al. 2019 Japan [67] |
| Rarely drinking | Reported alcohol consumption, but not in the past year | McCaul et al. 2010 Australia [113] |
| **MILD DRINKERS** | | |
| Mild drinking | > 1 standard drink/week | Kim et al. 2020 Korea [66] |
| Mild drinkers | < 20 g/day | Kohno et al. 2019 Japan [81] |
| **LIGHT TO MODERATE DRINKERS** | | |
| Very light drinkers | < 1 drink/week | Davis et al. 2014 Iceland [78] |
| Very light drinkers | < 14 g/day | Siddiquee et al. 2020 Japan [59] |
| Light drinkers | > 0 to < 3 drinks per week | Hoang et al. 2014 USA [60] |
| Light drinkers | $\leq$ 3 drinks/week | Moore et al. 2009 USA [115] |
| Light drinking | 0 < drinks $\leq$ 7 per week | Jeong et al. 2012 Korea [105] |

(*Continued*)

**Table 6.** (Continued)

| Category | Definition | Author and country |
|---|---|---|
| Light drinkers | 14–23 g/day | van Oort et al. 2020 The Netherlands [134] |
| Light drinkers | ≤ 1 standard drink/day and no binge drinking in the last 30 days | Soler-Vila et al. 2019 Spain [127] |
| Light drinkers | 0–30 g/week | van Oort et al. 2020 The Netherlands [134] |
| Light-to-moderate drinkers | 0.5–30 drinks/month | Agahi et al. 2016 Sweden [6] |
| Light-to-moderate drinkers | Women: 1–7 drinks/week | Davis et al. 2014 Iceland [78] |
| Light drinkers | Men: 1–7 drinks/week | Davis et al. 2014 Iceland [78] |
| Light/moderate alcohol consumption | ≤ 7 drinks/week, and ≤ 3 drinks/day | Wilson et al. 2014 USA [5] |
| Low-moderate drinking levels | 0–100 g alcohol/week | Waern et al. 2014 Sweden [12] |
| Light/moderate drinkers | ≤ 10 drinks/weeks, ≤ 4 drinks/day | Cousins et al. 2014 Ireland [90] |
| Light drinkers | Men: < 168 g alcohol/week; Women: < 112 g alcohol/week | Chan et al. 2010 China [62] |
| Light consumption | 1–2 drinks/day | Lima et al. 2009 Brazil [22] |
| Low-moderate drinking | Men: < 30 g/day, ≤ 140 g/week (> 0 and ≤ 14 units/week) Women: < 15 g/day in women, ≤ 70 g/week in women (> 0 and ≤ 7 units/week) | Bazal et al. 2019 Spain [75] |
| Light-to-moderate drinkers | > 30 - ≤ 70 g/week | van Oort et al. 2020 The Netherlands [134] |
| **REGULAR DRINKERS** | | |
| Regular drinkers | Dinking at least 1 drink/week on at least one occasion (over a period of 4 years) | Goulden et al. 2016 USA [1] |
| Regular drinkers | Alcohol consumption ≥ 1 day/week | Tevik et al. 2017 Norway [129] |
| Regular drinkers | 1–5 days/week | Kohno et al. 2019 Japan [81] |
| Regular drinkers | < 15 drinks/week | Almeida et al. 2014 Australia [55] |
| Non-risky alcohol intake | No daily alcohol consumption | Hajek et al. 2017 Germany [96] |
| **HABITUAL DRINKERS** | | |
| Habitual drinkers | 6–7 days/week | Kohno et al. 2019 Japan [81] |
| **LOW RISK DRINKNIG** | | |
| Low risk drinkers | ≤ 30 drinks during the month prior to the interview and ≤ 3 drinks/occasion | AlGhatrif et al. 2013 USA [15] |
| Low risk | Men: ≥ 1– < 30 g/day (up to 2 drinks a day); Women: ≥ 1 –< 15 g/day (up to 1 drink a day) | Dhana et al. 2020 USA [91] |
| Low risk | Men: ≤ 4 drinks/day; Women: ≤ 2 drinks/day (Australian guidelines) | Heffernan et al. 2016 Australia [99] |
| Low risk | Men: ≤ 4 drinks/day and ≤ 14 drinks/week; Women: ≤ 3 drinks/day and ≤ 7 drinks/week (US guidelines) | Heffernan et al. 2016 Australia [99] |
| Low alcohol intake | 4 drinks/week | Hassing 2018 Sweden [8] |
| Low risk | > 2 ≤ 2 drinks/day | Tait et al. 2013 Australia [128] |
| Low risk drinking | Men: ≤ 21 units/week; Women: ≤ 14 units/week | Iparraguirre et al. 2015 England [20] |
| Low risk drinkers | Men: ≤ 280 g alcohol/week or ≤ 28 standard drinks/week Women: ≤ 140 g alcohol/week or ≤ 14 standard drinks/week AUDIT-C score < 4 in men and < 3 in women | Roson et al. 2010 Spain [18] |
| **MEDITERRANEAN ALCOHOL DRINKING** | | |
| Mediterranean alcohol drinking | 10–30 g/day in men and 5–15 g/day in women, preferably red wine consumption with low spirits consumption | Bazal et al. 2019 Spain [75] |
| Mediterranean drinking pattern | < 40 g/day for men, < 24 g/day for women, no binge drinking, with preference for wine and drinking only with meals | Ortola et al. 2017 Spain [119] |
| **FREQUENT DRINKING** | | |
| Frequent drinkers | Alcohol consumption ≥ 4 days/week | Tevik et al. 2019 Norway [130] |
| Frequent drinking | Drinking 5–7 days/week | Wang et al. 2017 China [137] |

(*Continued*)

**Table 6.** (Continued)

| Category | Definition | Author and country |
|---|---|---|
| Frequent drinkers | $\geq$ 15 ($\geq$ 12 women) standard drinks in the entire week, but no more than 5 (4 women) on the same day | Nuevo et al. 2015 Europe [118] |
| Daily drinkers | Alcohol consumption on 7 days per week | Ilomaki et al. 2013 Australia [57], Ilomaki et al. 2014 Australia [58] |
| **MODERATE DRINKING** | | |
| Moderate users | $\geq$ 1 drink/month, but $\leq$ 7 drinks/week | Immonen et al. 2011 Finland [103] |
| Irregular moderate drinkers | < 1 drink/week | Halme et al. 2010 Finland [97] |
| Regular moderate drinkers | 1–7 drinks/week | Halme et al. 2010 Finland [97] |
| Moderate drinkers | 1–7 glasses/week | Hoeck et al. 2013 Belgium [100] |
| Moderate drinking | 1–7 units/week | Machado et al. 2017 Latin America [21] |
| Moderate drinkers | $\leq$ 3 drinks/day and $\leq$ 7 drinks/week | Scott et al. 2020 USA [80] |
| Moderate drinkers | $\leq$ 250 ml alcohol/day | Gargiulo et al. 2013 Italy [94] |
| Moderate drinkers | 0.5–1 standard drink per drinking episode | Hongthong et al. 2016 Thailand [102] |
| Moderate drinkers | $\geq$ 3 to $\leq$ 7 drinks per week | Hoang et al. 2014 USA [60] |
| Moderate drinking | $\leq$ 7 drinks/week | Ivan et al. 2014 USA [63] |
| Moderate drinkers | Men $\geq$ 65 years and women: $\leq$ 1 drink/day<br>Men < 65 years: $\leq$ 2 drinks/day | Richard et al. 2017 USA [4] |
| Moderate drinking | 8 drinks/week | Hassing 2018 Sweden [8] |
| Moderate drinkers | 1–14 units/week, 8–112 g/week (within United Kingdom guidelines) | Britton et al. 2020 United Kingdom [85] |
| Moderate drinkers | Men: 4–14 drinks/week; Women: 4–7 drinks/week | Moore et al. 2009 USA [115] |
| Moderate drinking | 7 < alcoholic drinks $\leq$ 14 per week | Jeong et al. 2012 Korea [105] |
| Moderate drinkers | Men: 7–14 drinks/week | Davis et al. 2014 Iceland [78] |
| Exceeding moderate drinking limits | > 1 drink/day for women and > 2 drinks/day for men (Centers for Disease Control and Prevention guidelines) | Sanford et al. 2020 USA [125] |
| Moderate drinkers | >70 - $\leq$ 140 g/week | van Oort et al. 2020 The Netherlands [134] |
| Moderate drinkers | 1–2 drinks/day | Towers et al. 2018 New Zealand [131] |
| Moderate occasionally | $\leq$ 2 units/day | Liu et al. 2019 Japan [111] |
| Moderate daily | $\leq$ 2 units/day | Liu et al. 2019 Japan [111] |
| Moderate drinking | Up to 2 drinks/day | Ilomaki et al. 2014 Australia [58] |
| Moderate drinkers | Men: 1–21 beverages/week; Women: 1–14 beverages/week | Heegaard et al. 2011 Denmark [16] |
| Moderate drinkers | 15–27 drinks/week | Almeida et al. 2014 Australia [55] |
| Moderate drinkers | Men: $\leq$ 30 g alcohol/day; Women: $\leq$ 20 g alcohol/day | Weyerer et al. 2009 Germany [139] |
| Moderate drinkers | 20–39.9 g/day | Kohno et al. 2019 Japan [81] |
| Moderate drinkers | > 23–46 g/day | Siddiquee et al. 2020 Japan [59] |
| Moderate drinkers | Men: < 40 g alcohol/day; Women: < 24 g alcohol/day | Ortola et al. 2017 Spain [119] |
| Moderate drinkers | Men: < 40 g alcohol/day; Women: < 25 g alcohol/day | Gonzàlez-Rubio et al. 2016 Spain [95] |
| Moderate consumption | 3–4 drinks/day | Lima et al. 2009 Brazil [22] |
| Moderate drinkers | Men: > 168 g, but < 400 g alcohol/week; Women: > 112 g, but < 280 g alcohol/week | Chan et al. 2010 China [62] |
| **ELEVATED ALCOHOL CONSUMPTION** | | |
| Elevated alcohol consumption | China: > 1 drink/day or > 7 drinks/week<br>Norway: Drinking 4–7 days a week | Li et al. 2019 China and Norway [109] |
| **RISK DRINKING** | | |
| Risk drinkers/at risk consumption | $\geq$ 8 units or drinks/week | Machado et al. 2017 Latin America [21], Sacco et al. 2009 USA [124] |
| Risky alcohol intake | Daily alcohol consumption | Hajek et al. 2017 Germany [96] |
| At risk drinkers | > 30 drinks during the month prior to the interview or > 3 drinks/occasion | AlGhatrif et al. 2013 USA [15] |

*(Continued)*

**Table 6.** (*Continued*)

| Category | Definition | Author and country |
|---|---|---|
| At-risk drinking | Consuming > 7 drinks/week or ≥ 5 drinks on a typical drinking day or using ≥ 3 drinks several times per week | Immonen et al. 2011 Finland [103], Immonen et al. 2013 Finland [104] |
| At risk drinkers/drinking | 8–14 glasses or drinks /week | Hoeck et al. 2013 Belgium [100], Ivan et al. 2014 USA [63] |
| At risk drinkers | > 1 standard drink (> 14 g of alcohol) per day or any binge drinking in the last 30 days (≥ 80 g for men and ≥ 60 g for women of alcohol in one session) | Soler-Vila et al. 2019 Spain [127] |
| At risk drinking | ≥ 100 g alcohol/week in men and women  Alternative cut-off for at-risk drinking in women: ≥ 60 g alcohol/week | Waern et al. 2014 Sweden [12] |
| At risk alcohol consumption | ≥ 100 g alcohol/week | Ahlner et al. 2018 Sweden [71] |
| Risky drinking | Men: ≥ 10 drinks/week; Women: ≥ 7 drinks/week; or ≥ 5 drinks at one sitting ≥ 1 time/year for both men and women | McClure et al. 2013 USA [2] |
| At risk drinking | Definition 1: >10 units of alcohol per week  Definition 2: >14 units of alcohol per week | Foster et al. 2019 England [93] |
| High risk | Men: ≥ 30 g/day; Women: ≥ 15 g/day | Dhana et al. 2020 USA [91] |
| Risky | Men: > 4 drinks/day; Women: > 2 drinks/day (Australian guidelines) | Heffernan et al. 2016 Australia [99] |
| Increased risk | Men: > 4 drinks/day or > 14 drinks/week; Women: > 3 drinks/day or > 7 drinks/week (US guidelines) | Heffernan et al. 2016 Australia [99] |
| Risky alcohol consumption assessed with use of SIAC | Men: > 280 g alcohol/week or > 28 standard drinks/week; Women: > 140 g alcohol/week or > 14 standard drinks/week | Roson et al. 2010 Spain [18] |
| Increased risk drinking | Men: 22 ≤ 50 units/week; Women: 15 ≤ 35 units/week | Iparraguirre 2015 England [20] |
| At-risk drinkers | Men: > 30 g alcohol/day; Women: > 20 g alcohol/day | Weyerer et al. 2009 Germany [139] |
| Long term risk | > 2 ≤ 4 drinks/day | Tait et al. 2013 Australia [128] |
| Short term risk | > 4 drinks/day | Tait et al. 2013 Australia [128] |
| Unsafe drinkers | Men: > 21 units/week; Women: > 14 units/week | Rao et al. 2015 United Kingdom [122] |
| Highest risk | Men: both > 4 drinks/day and > 14 drinks/week; Women: both > 3 drinks/day and > 7 drinks/week (US guidelines) | Heffernan et al. 2016 Australia [99] |
| Higher risk drinking | Men: > 50 alcohol units/week; Women: > 35 units/week | Iparraguirre et al. 2015 England [20] |
| **UNSAFE DRINKING** | | |
| Unsafe drinking | ≥ 14 standard drinks/week, ≥ 140 g/week | Kim et al. 2020 Korea [66] |
| **HEAVY DRINKING** | | |
| Heavy drinkers | > 30 drinks/month | Agahi et al. 2016 Sweden [6], Shaw et al. 2011 USA [126] |
| Heavy drinkers | ≥ 8 drinks/week | Hoang et al. 2014 USA [60] |
| Heavy drinking | ≥ 8 drinks (≥ 96 g alcohol) in a week or ≥ 4 drinks (≥ 48 g alcohol) at least in one day last 28 days | Aalto et al. 2011 Finland [50] |
| Heavy drinkers | 8–14 drinks/week | Halme et al. 2010 Finland [97] |
| Heavy occasionally | > 2 units/day | Liu et al. 2019 Japan [111] |
| Heavy daily | > 2 units/day | Liu et al. 2019 Japan [111] |
| Heavy drinkers | >140 g/week | van Oort et al. 2020 The Netherlands [134] |
| Heavy drinking | Men: ≥ 30 g/day, > 140 g/week (> 14 units/week); Women: ≥ 15 g/day, > 70 g/week (>7 units/week) | Bazal et al. 2019 Spain [75] |
| Heavy drinkers | > 10 drinks/week, > 4 drinks/day | Cousins et al. 2014 Ireland [90] |
| Heavy drinkers | Men: > 14 drinks/week; Women: > 7 drinks/week; and > 7 drinks/week for women and men | Moore et al. 2009 USA [115] |
| Heavy drinkers | Men: > 14 drinks/week; Women: > 7 drinks/week | Davis et al. 2014 Iceland [78] |
| Heavy drinking | > 14 drinks/week | Ivan et al. 2014 USA [63], Jeong et al. 2012 Korea [105] |
| Heavy drinkers | > 2 drinks/day | Ilomaki et al. 2013 Australia [57] |
| Heavy drinkers | ≥ 15 units/week (above United Kingdom guidelines) | Britton et al. 2020 United Kingdom [85] |

(*Continued*)

**Table 6.** (*Continued*)

| Category | Definition | Author and country |
|---|---|---|
| Heavy drinkers | Men ≥ 65 years and women: > 1–3 drinks/day | Richard et al. 2017 USA [4] |
| Heavy drinkers | ≥ 15 standard drink units (≥ 12 women) during the week, and ≥ 5 (≥ 4 women) on at least one day | Nuevo et al. 2015 Europe [118] |
| Heavy drinkers | ≥ 15 drinks/week | Halme et al. 2010 Finland [97] |
| Heavy drinkers | 15–21 glasses/week | Hoeck et al. 2013 Belgium [100] |
| Heavy drinkers | Men: > 17 standard drinks/week; Women: > 11 standard drinks/week; Men/women: ≥ 6 standard drinks per drinking occasion | Holton et al. 2019 Ireland [101] |
| Heavy drinkers | Men: > 21 beverages or units/week; Women: > 14 beverages or units/week | Heegard et al. 2011 Denmark [16], Nadkarni et al. 2011 Dominican Republic [116] |
| Heavy drinkers | Men: ≥ 40 g alcohol/day; Women: ≥ 24 g alcohol/day | Ortola et al. 2017 Spain [119] |
| Heavy drinkers | ≥ 40 g/day | Kohno et al. 2019 Japan [81] |
| Heavy drinkers | > 46 g/day | Siddiquee et al. 2020 Japan [59] |
| Heavy drinkers | Men: > 400 g alcohol/week; Women: > 280 g alcohol/week | Chan et al. 2010 China [62] |
| Heavy drinkers | ≥ 3 drinks/day | Towers et al. 2018 New Zealand [131] |
| Heavy drinking | 3–4 drinks/day | Ilomaki et al. 2014 Australia [58] |
| Heavy consumption | ≥ 5 drinks/day | Lima et al. 2009 Brazil [22] |
| **PROBLEMATIC/HARMFUL DRINKING** | | |
| Problematic alcohol use | > 1 standard drink/day or > 7 standard drinks/week and > 3 drinks on one occasion | Aguila et al. 2016 Mexico [82] |
| Problematic drinkers | > 21 glasses/week | Hoeck et al. 2013 Belgium [100] |
| Harmful drinking | Men: > 60 g alcohol/day; Women: > 40 g alcohol/day | Weyerer et al. 2011 Germany [138] |
| **EXCESSIVE DRINKING** | | |
| Excessive alcohol consumption | > 1 unit/day at a sitting or > 7 units/week | Lasebikan et al. 2015 Nigeria [107] |
| Excessive drinkers | > 3 drinks/day for men ≥ 65 years and women, > 4 drinks/day for men < 65 years | Richard et al. 2017 USA [4] |
| Excessive drinking | > 4 drinks/day | Ilomaki et al. 2014 Australia [58] |
| Excessive drinking | ≥ 6 alcoholic drinks per occasion last 3 months | Listabarth et al. 2020 12 European countries [110] |
| Regular excessive alcohol consumption | > 6 drinks/day | Almeida et al. 2017 Australia [56] |
| Extremely high consumption | Men: > 120 g alcohol/day; Women: > 80 g alcohol/day | Weyerer et al. 2011 Germany [138] |
| **BINGE DRINKING/HEAVY EPISODIC DRINKING** | | |
| Binge drinking | Men: ≥ 4 drinks per drinking day; Women: ≥ 3 drinks per drinking day | Choi et al. 2011 USA [140] |
| Binge drinking | > 4 drinks on one occasion | Hoang et al. 2014 USA [60], Villalonga-Olives et al. 2020 USA [135] |
| Binge drinking | ≥ 5 drinks for men, ≥ 4 drinks for women on at least one occasion (at the same time or within a couple of hours apart) in the past 30 days | Parikh et al. 2015 USA [121] |
| Binge drinking | Men: ≥ 5 drinks on the same occasion; Women: ≥ 4 drinks on the same occasion. | Han et al. 2019 USA [98] |
| Past year binge drinking | Men: ≥ 5 drinks in one day/sitting; Women: ≥ 4 drinks in one day/sitting | Bryant et al. 2013 USA [86], Bryant et al. 2013 USA [87] |
| Binge drinking | ≥ 5 drinks at least once per month | Ilomaki et al. 2013 Australia [57], Ilomaki et al. 2014 Australia [58] |
| Binge drinking | ≥ 5 drinks on the same occasion in at least one day in the past 30 days | Marti et al. 2015 USA [112] |
| Binge drinking | ≥ 5 drinks on any occasion within the past 30 days | Davis et al. 2014 Iceland [78] |
| Binge drinking | ≥ 5 drinks during at least one day over the past year in women and men | Sanford et al. 2020 USA [125] |
| Binge drinking | ≥ 5 drinks in at least one day in the past 12 months | Wilson et al. 2014 USA [5] |
| Binge drinking | ≥ 6 standard drinks on one occasion at least monthly | Fuentes et al. 2017 Europe [7], Jeong et al. 2012 Korea [105] |

(*Continued*)

**Table 6.** (Continued)

| Category | Definition | Author and country |
|---|---|---|
| Binge drinking | > 6 standard drinks per drinking day past year | Kim et al. 2020 Korea [66] |
| Binge drinking | ≥ 80 g in men, ≥ 60 g in women, during any drinking session in the preceding 30 days | Ortola et al. 2017 Spain [119] |
| Heavy episodic drinking | ≥ 4 drinks in a single day in a typical month last year | Merrick et al. 2011 USA [114] |
| Heavy episodic drinking | ≥ 4 drinks in any single day | Ryan et al. 2013 USA [123] |
| Heavy occasional drinkers | ≥ 5 (≥ 4 in women) standard drinks in one day but no more than 15 (12 in women) standard drinks in the entire week | Nuevo et al. 2015 Europe [118] |
| Heavy episodic use | ≥ 5 drinks on one occasion in the past year | Sacco et al. 2009 USA [124] |
| Heavy episodic drinking | ≥ 5 (5–7 or ≥ 8) on ≥ one occasions in the prior year | Satre et al. 2011 USA [64] |
| **SCREENING TOOLS** | | |
| AUDIT-C: hazardous drinking | AUDIT-C score ≥ 4 in men and ≥ 3 in women | Bell et al. 2015 England [84] Towers et al. 2019 New Zealand [132] |
| AUDIT-C: elevated alcohol consumption | AUDIT-C score ≥ 4 in men and ≥ 3 in women | Johannessen et al. 2017 Norway [19] |
| AUDIT-C: increased risk of hazardous drinking/alcohol abuse or dependence | AUDIT-C score ≥ 4 in men and ≥ 3 in women | Roson et al. 2010 Spain [18] |
| AUDIT-C: heavy drinking | AUDIT cutoff ≥ 4 (sensitivity 94% and specificity 80%) | Aalto et al. 2011 Finland [50] |
| AUDIT-C: high risk alcohol consumption | AUDIT-C score ≥ 4 | Jentsch et al. 2017 Germany [61] |
| AUDIT-C: hazardous drinking | AUDIT-C score ≥ 5 | Britton et al. 2020 United Kingdom [85] |
| AUDIT-C: | Low-risk drinking: Men: AUDIT-C score 1–3; Women: AUDIT-C score 1–2 <br> Moderate-risk drinking: Men: AUDIT-C score 4–7; Women: AUDIT-C score 3–7 <br> High-risk drinking: Men/women: AUDIT-C score 8–12 | Chavez et al. 2016 USA [73] |
| AUDIT: heavy drinking | AUDIT cutoff ≥ 5 (sensitivity 86% and specificity 87%) | Aalto et al. 2011 Finland [50] |
| AUDIT | Low risk use: AUDIT score 0–7 <br> Risk use: AUDIT score 8–14 | Villar Luis et al. 2018 Brazil [136] |
| AUDIT: hazardous and harmful drinking | AUDIT-score ≥ 8 in men and ≥ 6 in women | Roson et al. 2010 Spain [18] |
| AUDIT: at-risk drinking | AUDIT score ≥ 8 | Jeong et al. 2012 Korea [105] |
| AUDIT: problem drinkers | AUDIT score ≥ 12 | Kim et al. 2015 Korea [106] |
| AUDIT | Low risk: AUDIT score 0–7 <br> Risky alcohol use: AUDIT score 8–12 <br> High risk alcohol use: AUDIT score ≥ 13 <br> High consumption risk: AUDIT score ≥ 6 for question 1–3 may indicate risk of alcohol-related harm <br> High dependence risk: AUDIT score ≥ 4 for question 4–6, possibility of alcohol dependence <br> High alcohol-related problems risk: any score > 0 for question 7–10 | Vafeas et al. 2017 Australia [133] |
| CAGE: problem drinking | CAGE score ≥ 2 | Cousins et al. 2014 Ireland [90] |
| CAGE: alcohol problem | CAGE score ≥ 2 or CAGE 65+ score ≥ 1 | Hoeck et al. 2013 Belgium [100] |
| CAGE: problem drinkers | CAGE score ≥ 2 | Ilomaki et al. 2013 Australia [57], Ilomaki et al. 2014 Australia [58] |
| CARET: at risk drinking | Categorized as 1) Alcohol use behaviors in the last 12 months, 2) Alcohol use and medications taken at least 3–4 times per week currently, 3) Alcohol use and comorbidities in the past 12 months | Barnes et al. 2010 USA [83] |
| CARET: hazardous drinking | 27-item. Evaluates hazardous drinking regarding to level of alcohol use (frequency, quantity, and binge) and whether such drinking occurs in the presence of critical factors known to increase the risk of alcohol-related harm for older adults (i.e., comorbidities, use of alcohol-interactive medication [e.g., analgesics], and alcohol risk behaviors [e.g., driving after drinking alcohol]). | Towers et al. 2019 New Zealand [132] |
| MAST-G: alcohol related problems | MAST-G score ≥ 5 | Villar Luis et al. 2018 Brazil [136] |
| SMAST: possible lifetime alcohol problems | SMAST score ≥ 3 | Satre et al. 2011 USA [64] |
| **ALCOHOL ABUSE, DEPENDENCE OR MISUSE** | | |

*(Continued)*

**Table 6.** (*Continued*)

| Category | Definition | Author and country |
|---|---|---|
| Current alcohol abuse | ≥ 1 of 4 criteria last 12 months obtained via the Mini International neuropsychiatric Interview | Guidolin et al. 2016 Brazil [72] |
| Current alcohol dependence | ≥ 3 of 7 criteria last 12 months obtained via the Mini International neuropsychiatric Interview | Guidolin et al. 2016 Brazil [72] |
| Abuse | ≥ 35 drinks/week | Almeida et al. 2014 Australia [55] |
| Lifetime alcohol misuse | Score ≥ 1 from the five-item Self-Reporting Questionnaire | Nogueira et al. 2013 Brazil [117] |
| Major lifetime alcohol misuse | Score ≥ 2 from the five-item Self-Reporting Questionnaire | Nogueira et al. 2013 Brazil [117] |
| **DSM-IV** | | |
| DSM-IV: Alcohol abuse or alcohol dependence | | Jeong et al. 2012 Korea [105] |
| DSM-IV: Abuse | Current, 12-month, and lifetime | Muñoz et al. 2018 Europe [9] |
| DSM-IV: Dependence | Current, 12-month, and lifetime | Muñoz et al. 2018 Europe [9] |
| DSM-IV: Alcohol use disorder | Current, 12 month, and lifetime | Muñoz et al. 2018 Europe [9] |
| DSM-IV: Past year alcohol abuse or dependence | | Sacco et al. 2009 USA [124] |
| **ACCORDING TO GUIDELINES** | | |
| Within-guidelines drinkers | ≤ 30 drinks/month, or ≤ 3 drinks on a single day | Merrick et al. 2011 USA [114] |
| Within guidelines drinkers | Not exceeding the monthly limit (≤ 30 drinks per typical month) or the single day limit (< 4 drinks in any single day) | Ryan et al. 2013 USA [123] |
| Exceeding monthly limits | > 30 drinks per typical month | Ryan et al. 2013 USA [123] |
| Exceeding the monthly limit, but not single day limit | > 30 drinks/months, ≤ 3 drinks on a single day | Merrick et al. 2011 USA [114] |
| Exceeded guidelines: At risk drinkers | Exceeded the monthly limit (> 30 drinks per typical month), but not the single-day limit (< 4 drinks in any single day), and heavy episodic drinkers who exceeded the single-day drinking limit (≥ 4 drinks) with or without exceeding the monthly limit. | Ryan et al. 2013 USA [123] |
| Drinking in excess of NIAAA guidelines | > 1 drink/day for women of any age and men ≥ 65 years; > 2 drinks/day for men < 65 years | McEvoy et al. 2013 USA [3] |
| Alcohol in excess of NIAAA limits | > 7 drinks/week, > 3 drinks/day | Wilson et al. 2014 USA [5] |
| **FREQUENCY** | | |
| Frequency of alcohol consumption | Non-drinkers; Infrequent drinkers: < 2 times/month; Weekly: 1–4 times/week; Near daily drinkers: 5–7 times/week | Richard et al. 2017 USA [4] |
| Frequency of alcohol consumption | Never; ≤ 1 time a month; 2–4 times a month; 2–3 times a week; 4–5 times a week; daily/almost daily | Wilson et al. 2014 USA [5] |
| Frequency of alcohol consumption | Monthly or less; Every week: 1–4 times/week; Daily or almost daily: 5–7 times/week | Agahi et al. 2019 Sweden [74] |
| Weekly drinking frequency | ≥ 2 days/week, ≥ 3 drinks/week | Aguila et al. 2016 Mexico [82] |
| Number of drinking days/week | 0 days (non-drinkers), 1–7 days | Choi et al. 2011 USA [140] |
| **QUANTITY** | | |
| g alcohol/day | Volume consumed in men: ≤ 12, 13–24, 25–47, ≥ 48 g/day<br>Volume consumed in women: ≤ 12, 13–24, > 24 g/day | Buja et al. 2010 Italy [77], Buja et al. 2011 Italy [76] |
| Daily alcohol consumption | 20 g/day; 21–59 g/day; ≥ 60 g/day | Tateishi et al. 2019 Japan [69] |
| g alcohol/week | 0, 1–20, 20–40, 40–60, 100–150, 150–250, 250–500 or > 500 g alcohol/week | Ahlner et al. 2018 Sweden [71] |
| Alcohol units per week | None in the past week, 1–14 units, 14–21 unit, > 21 units | Britton et al. 2020 United Kingdom [85] |
| Alcohol units last week | Categorization 1:<br>None (last 12 months); < 1 unit; 1–7 units; 7–10 units; > 10–14 units; > 14–21 units; > 21–28 units; > 28–35 units; > 35–50 units; > 50 units<br>Categorization 2:<br>None or < 1 unit last 12 months; 1–10 units; > 10–21 units; > 21–28 units; > 28–50 units; > 50 units | Foster et al. 2019 England [93] |
| Drinks/day | ≤ 1, 2, 3, 4, ≥ 5 drinks/day | Wilson et al. 2014 USA [5] |

(*Continued*)

**Table 6.** (Continued)

| Category | Definition | Author and country |
|---|---|---|
| Drinks/day | ≤ 2 drinks/day, 2–4 drinks/day, 4–6 drinks/day, > 6 drinks/day | Almeida et al. 2017 Australia [56] |
| Drinks/month | Mean monthly drinks | Bryant et al. 2019 USA [88] |
| Number of drinks on drinking days | Men: 0–4 drinks/day; Women: 0–3 drinks/day | Choi et al. 2011 USA [140] |
| Lifetime cumulative dose of alcohol | Quartiles of lifetime total units of alcoholic beverages | D'Ovidio et al. 2019 Ireland, the Netherlands, and Italy [65] |
| **ALCOHOL CONSUMPTION BY YEAR** | | |
| Duration of drinking | Never; 0–19 year; 20–39 year; ≥ 40 years | Zaitsu et al. 2020 Japan [70] |
| Cumulative drinking amount | Categorization according to g/day-years: Never; < 1000; 1000–2250; 2250–4000; ≥ 4000 | Suo et al. 2019 China [68] |
| Drink-year levels | Classification 1: 0 (lifetime abstainer = never consumed alcohol); > 0–20 drink-years; > 20–40 drink-years; > 40–60 drink-years; > 60–90 drink-years; > 90 drink-years Classification 2: 0 drinks per day (lifetime abstainer); ≤ 2 drinks/day and < 20 years; ≤ 2 drinks/day and 20–39 years; ≤ 2 drinks/day and ≥ 40 years; > 2 drinks/day and < 20 years; > 2 drinks/day and 20–39 years; > 2 drinks/day and ≥ 40 year | Zaitsu et al. 2020 Japan [70] |

g = grams.

AUDIT = Alcohol Use Disorders Identification Test; AUDIT-C = Alcohol Use Disorders Identification Test, short form; CAGE = Cut down, Annoyed, Guilty, Eye opener; CARET = Comorbidity Alcohol Risk Evaluation; DSM-IV = Diagnostic and Statistical Manual of Mental Disorders, 4th Edition; ICD-10 = International Classification of Diseases 10th Revision; MAST-G = Michigan Alcoholism Screening Test-Geriatric Version; NIAAA = National Institute on Alcohol Abuse and Alcoholism; SIAC = Systematic Inventory Alcohol questionnaire; SMAST = Short Michigan Alcohol Screening Test.

## Recall period

Several studies (N = 34) included in this review used a recall period of the last 12 months, which is in line with the WHO's recommendation [24]. It is suggested that a recall period of the last 12 months will give the most valid assessment of alcohol consumption [24, 26, 43]. However, 28 studies used a recall period of the last month or last week. A shorter recall period minimizes problems of memory loss [45], which is important when studying alcohol consumption in older adults. Older adults might find it difficult to recall their alcohol consumption in the last 12 months [43], and a shorter recall period could yield a more reliable assessment of alcohol consumption [24]. However, a recall period of the last week or last month might not represent older adults' typical drinking pattern in the last year, as older adults might be irregular drinkers and might not have been drinking during the last month [43]. Consequently, older adults with an infrequent drinking pattern might be misclassified as abstainers with the use of a short recall period [44, 45]. Infrequent heavy episodic drinkers could also be wrongly classified with the use of a short recall period [24]. Moreover, in studies assessing alcohol-related problems, it is particularly important to use a long recall period, as alcohol-related problems only can be measured with sufficient precision over a period of at least one year [25, 45]. Thus, a recall period of the last 12 months seems desirable in studies assessing alcohol consumption in older adults.

However, an unexpected high number of the included studies (N = 35) did not report the recall period at all. Missing information regarding the recall period will complicate the interpretation of the importance of alcohol consumption for health and well-being in older adults.

## Measure of alcohol consumption

Most of the studies (N = 67) reported the total volume of alcohol consumption in standard drinks, units, or glasses, whereas 27 studies reported the volume of alcohol consumption in

grams. Presenting the volume in grams might be difficult for the reader to interpret, whereas presenting the results in drinks could be problematic, as drinkers frequently do not consume standard drinks. In addition, many are not familiar with the concept of standard drinks, which makes it difficult to estimate the number of consumed drinks [24, 25, 27]. Participants are likely to report drink sizes they actually consume which differ from the size of standard drinks [26]. Thus, as already mentioned, pictures of a standard drink of beer, wine, or liquor could be helpful for older adults when estimating how much they have been drinking [24, 26]. This method was used by Nuevo and colleagues [118], who examined older adults' drinking patterns in 14 European countries. The interviewers showed the participants a card with pictures representing one drink of alcohol according to the standard for each country [118].

In this systematic review, the definition of one standard drink varied from 8 grams of alcohol to 50 grams of alcohol. Thus, using the term drinks in alcohol surveys will complicate international comparisons [24]. The question has been raised of whether an international universal standard of drinks should be established [24]. In the meantime, the WHO suggests that alcohol consumption should be reported in grams of alcohol for the sake of international comparison [24].

## Strengths and limitations

The strengths of our review include the use of five widely recognized bibliographic databases: MEDLINE, PubMed, CINAHL, PsycINFO, and EMBASE.

One limitation of this review is the exclusion of studies published in a language other than English and older studies published before 2009. Thus, there could be studies written in other languages and older studies that used other definitions that are not reported in this review. Even so, this review detected a wide range of definitions of different drinking patterns.

In alcohol surveys, it is recommended to ask about the drinking context, which focuses on drinking with meals or not, drinking alone or not (e.g., with family members, friends, work colleagues, etc.), drinking on a weekday or on a weekend, and drinking in public (bars and restaurants) or at home [24, 25, 43]. The drinking context seems to be an important factor explaining alcohol consumption and the risk of alcohol consumption [43]. However, in this systematic review, we did not assess drinking context in the included studies. It is desirable that coming studies ask about drinking context when assessing alcohol consumption in older adults [24, 43].

## Implications

We want to acknowledge the previous expert groups and alcohol epidemiologists for their effort to standardize the alcohol methodology in adult general population surveys [24, 25, 40, 43, 44]. However, it seems thus far that they have failed to fully achieve a standardization, and especially for the subgroup of the aged population. Different aims, traditions, and simple research group preferences may have resulted in the variety of measures and definitions found in this systematic review. Future research should work toward establishing a standardized assessment and definition of drinking patterns in older adults, especially risk drinking and heavy episodic drinking. Methodological studies are needed to study the reliability and validity of different assessment instrument and definitions [46]. Standardized assessments and definitions will contribute to improving the comparison of findings between studies and countries and to drawing firm conclusions about the prevalence and health effect of different drinking patterns. Use of a standardized and concise methodology in alcohol surveys of older adults could also lead to more informed and evidence-based policymaking to reduce alcohol's burden on health and economy [43].

## Conclusions

Several previous expert groups and alcohol epidemiologists have had an aim of standardizing the alcohol methodology in adult general population surveys. However, so far it seems that they have failed to fully achieve a standardization, and especially in the subgroup of the aged population. This systematic review shows that the included studies (N = 105) varied widely in the questionnaire applied, definitions, and measures to define drinking patterns in older adults. Different aims, traditions, and simple research group preferences may have resulted in the variety of measures and definitions found in this systematic review. Identical assessments and definitions need to be developed and used to make valid comparisons of alcohol consumption in older adults. In total, we detected 19 different drinking patterns, and each drinking pattern had a wide range of definitions. We recommend that alcohol surveys in older adults define the following drinking patterns: lifetime abstainers, former drinkers, current drinkers, risk drinking, and heavy episodic drinking. The definitions used by the WHO for lifetime abstainers, former drinkers, and current drinkers are recommended to be used for older adults. Standardized and valid definitions of risk drinking and heavy episodic drinking should be developed. The expanded QF questionnaire including three questions with a focus on drinking frequency, drinking volume, and heavy episodic drinking, with a recall period of 12 months, could be used.

## Supporting information

**S1 Checklist. PRISMA 2009 checklist.**
(DOC)

**S1 Table. Self-report measures used in epidemiological studies to assess alcohol consumption among older adults.**
(DOCX)

## Acknowledgments

We would like to acknowledge Vigdis Knutsen, the librarian at the Norwegian National Advisory Unit on Ageing and Health, who developed and executed the search strategy.

## Author Contributions

**Formal analysis:** Kjerstin Tevik, Anne-S. Helvik.

**Methodology:** Kjerstin Tevik, Sverre Bergh, Geir Selbæk, Aud Johannessen, Anne-S. Helvik.

**Writing – original draft:** Kjerstin Tevik, Sverre Bergh, Geir Selbæk, Aud Johannessen, Anne-S. Helvik.

**Writing – review & editing:** Kjerstin Tevik, Sverre Bergh, Geir Selbæk, Aud Johannessen, Anne-S. Helvik.

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
