## [Decision Letter · Decision Letter 0]

19 Oct 2021

PONE-D-21-29079Various self-report measures used in epidemiological studies to assess alcohol consumption among older adults – A systematic reviewPLOS ONE

Dear Dr. Tevik,

Thank you for submitting your manuscript to PLOS ONE. After careful consideration, we feel that it has merit but does not fully meet PLOS ONE’s publication criteria as it currently stands. Therefore, we invite you to submit a revised version of the manuscript that addresses the points raised during the review process. The revised version should address all concerns.

We look forward to receiving your revised manuscript.

Kind regards,

Petri Böckerman

Academic Editor

PLOS ONE

Journal Requirements:

Reviewers' comments:

Reviewer's Responses to Questions

**Comments to the Author**

1. Is the manuscript technically sound, and do the data support the conclusions?

Reviewer #1: Yes

Reviewer #2: Yes

2. Has the statistical analysis been performed appropriately and rigorously? 

Reviewer #1: Yes

Reviewer #2: Yes

3. Have the authors made all data underlying the findings in their manuscript fully available?

Reviewer #1: Yes

Reviewer #2: Yes

4. Is the manuscript presented in an intelligible fashion and written in standard English?

Reviewer #1: Yes

Reviewer #2: Yes

5. Review Comments to the Author

Reviewer #1: This is an important systematic review of alcohol consumption and pattern measurement for epidemiological studies of older adults (65 plus years old). The review including 105 studies winnowed from many more in a comprehensive search found wide variation in every aspect of measurement strategy and importantly alcohol use pattern definitions. Basic variance was also found in drink size included, duration of measurement and type of measure, i.e., QF, BSQF, GQF Weekly diary, Last occasion. Conclusions and implications proposed by the authors emphasized the need for standardization. The manuscript is very well written and provides a wealth of tabular material.

I have a number of comments and suggestions for improvement of the paper:

1) LL 104-113 The list of measurement approaches is fine but there is here and in many places following an overuse of secondary references 24 and 36, (WHO international measurement guidelines and SMART, respectively). While I have nothing against these citations based essentially of expert groups, some other citations could help reduce the over-reliance one example being Greenfield & Kerr, Alcohol measurement methodology in epidemiology: recent advances and opportunities- Addiction, 2008. It would be worth emphasizing that last drinking occasion (and yesterday drinking especially) typically have specialized uses for better calibrating longer duration measures such as 12-month measures (e.g., Stockwell et al. What did you drink yesterday? Etc. Addiction, 2008).

2) I commend the authors for identifying LL 59-64 identifying perennial problems of measurement and pattern definition. A type of pattern definition that does not appear to have been given sufficient attention in the excellent review (perhaps because the older age literature has attended to it less than in general population studies), is that of relative binging vs spacing of drinks to obtain the same volume. At the lower weekly levels such as those recommended for older adults by NIAAA 5/4+ (M/W) drinks or (5 or 4 x 12 g = 60 or 40 g ethanol) is a good indicator of this intake variability. At the higher volume levels 8+ may separate spacers (or even drinkers) from bunched (or binge) drinking patterns (per the GQF measure used, however, in only one of the included studies: Satre et al. [53]). Some recommendation for a smaller set of items have noted the value of including maximum in any day (usually categorized).

3) LL 133-137 The lack of standardization – aim of a systematic review somewhat puts the cart before the hose, in the sense that the ultimate finding that there is very little standardization seems to motivate the review that finds this. The conclusions and Implications sections echo the hope for moving toward more standardization, but I think it might be acknowledges somewhere that different aims (as well as traditions and simple research group preferences) result in the variety of measures and definitions seen. The authors could note (relevant although not specific to assessing drinking patterns among older adults) that over the years there have been expert meetings and groups convened to discuss alcohol measurement and patterns, and sometimes to make some consensus recommendations. One meeting organized by Stockwell was in Freemantle AU, and led to reference 24. Another was at NIAAA (See Alcohol Clin Exp Res 22, 4S-14S & 52S-62S, 1998). A third was a conference organized by Room held at Skarpö, Sweden in 2000, resulting in a special edition of the J Substance Use (e.g., Dawson & Room, Toward agreement on ways to measure and report drinking patterns and alcohol-related problems in adult general population surveys: the Skarpö Conference overview. J Subst Abuse 12, 1-21, 2000. SMART [Ref 36] represents a more recent effort. I therefore feel that the conclusions regarding standardization might be tempered by acknowledging the many previous such efforts of groups of alcohol epidemiologists, which have thus far failed to fully achieve such a standardization of alcohol measures and pattern definitions for general population surveys and similarly by extension for the aging subgroup. However, the authors are commended for recognizing that patterns and guidelines could differ in the elder group, and that on guidelines there is no empirically backed international guideline (LL 542-543).

4) LL 483-484 An important reference that complements [134] is Rehm, et al. Assessment methods for alcohol consumption, prevalence of high risk drinking and harm: a sensitivity analysis. Int J Epidemiol 28, 219-224, 1999. Even though GQF was only used in one paper in the review, this proved superior to QF and weekly diary in capturing risky and harmful drinking volumes.

Reviewer #2: Dr. Tevik and collaborators reported in an elegant systematic review how different are the parameters used for the categorization/assessment of alcohol drinking among epidemiological studies. This paper raised an important question since the assessment of alcohol consumption is used for the diagnostic of alcohol use disorder, thus if studies/hospitals don’t use the same criteria and definitions the diagnostics could be under or overestimated. All the criteria used for the studies selection/inclusion were well thinking and organized. Authors also followed a recommended structure for systematic review (prism9). Data is well reported and clear in all tables throughout the manuscript. All important “side” points were mentioned, such as problems with stigma; underestimation of alcohol consumption during self-report; the population’s knowledge about the definitions of standard-drinking and the use of figures to clarify that during interviews. The limitations and strengths were also mentioned. My two suggestions are: 1) Flip the title: A systematic review of self-report measures used in epidemiological studies to assess alcohol consumption among older adults; 2) Although authors included the country of each study on table 3, would be also interesting if they keep that information on table 6, so the readers would have an idea about which are the criteria used in each country. Besides that, the study is timely, the methodology applied is sound, the authors provide clear models and hypothesis, and the findings are clearly embedded in a theoretical framework in the discussion section.

6. PLOS authors have the option to publish the peer review history of their article (what does this mean?). If published, this will include your full peer review and any attached files.

Reviewer #1: No

Reviewer #2: No

---

## [Author Response · Author response to Decision Letter 0]

17 Nov 2021

Academic Editor PLOS ONE

Petri Böckerman 

Dear Dr Petri Böckerman,

Please find enclosed a copy of the manuscript “A systematic review of self-report measures used in epidemiological studies to assess alcohol consumption among older adults”. The article has not been published or submitted elsewhere. The authors have contributed to the conception, conduct and analyses of the study, and preparation of the manuscript according to the requirements of your journal and the Vancouver conversion, with no research misconduct. 

We appreciated the review. The revision has now been carried out. The entire document with the review from the reviewers is copied below, and the response of the review is written in red. We are looking forward to receiving a response from you regarding our manuscript. 

Sincerely yours,

Kjerstin Tevik 

Sverre Bergh Geir Selbæk Aud Johannessen Anne-S. Helvik 

Corresponding author

Kjerstin Tevik

Holemslykkjvegen 63

7224 Melhus

Telephone: +47 93039386

e-mail: kjerstin.e.tevik@ntnu.no

PONE-D-21-29079

Various self-report measures used in epidemiological studies to assess alcohol consumption among older adults – A systematic review

PLOS ONE

Dear Dr. Tevik,

Thank you for submitting your manuscript to PLOS ONE. After careful consideration, we feel that it has merit but does not fully meet PLOS ONE’s publication criteria as it currently stands. Therefore, we invite you to submit a revised version of the manuscript that addresses the points raised during the review process.

The revised version should address all concerns.0

The authors of the present systematic literature review agreed regarding methodological strategy, including for example inclusion/exclusion criteria and definitions and wrote the agreement down (in Norwegian) prior to study start. However, we do not have a fully written protocol in English, and thus, no protocol is attached to this resubmission.

We look forward to receiving your revised manuscript.

Kind regards,

Petri Böckerman

Academic Editor

PLOS ONE

Journal Requirements:

Unfortunately, we do not have a minimal data set underlying the results described in our manuscript. The results presented in this systematic review are found in the six tables in the main manuscript and in the large supplemental table. 

The reference list is reviewed to ensure that it is correct. 

The reference list is changed. The following new references are included in the manuscript and in the reference list:

Azagba, S., Shan, L., Latham, K., & Manzione, L. (2020). Trends in Binge and Heavy Drinking among Adults in the United States, 2011-2017. Subst Use Misuse, 55(6), 990-997. doi:10.1080/10826084.2020.1717538

Dawson, D. A., & Room, R. (2000). Towards agreement on ways to measure and report drinking patterns and alcohol-related problems in adult general population surveys: the Skarpö conference overview. J Subst Abuse, 12(1-2), 1-21. doi:10.1016/s0899-3289(00)00037-7

Greenfield, T. K. (1998). Evaluating competing models of alcohol-related harm. Alcohol Clin Exp Res, 22(2 Suppl), 52s-62s. doi:10.1097/00000374-199802001-00008

Greenfield, T. K., & Kerr, W. C. (2008). Alcohol measurement methodology in epidemiology: recent advances and opportunities. Addiction, 103(7), 1082-1099. doi:10.1111/j.1360-0443.2008.02197.x

Liu, W., Redmond, E. M., Morrow, D., & Cullen, J. P. (2011). Differential effects of daily-moderate versus weekend-binge alcohol consumption on atherosclerotic plaque development in mice. Atherosclerosis, 219(2), 448-454. doi:10.1016/j.atherosclerosis.2011.08.034

Midanik, L. T. (1994). Comparing usual quantity/frequency and graduated frequency scales to assess yearly alcohol consumption: results from the 1990 US National Alcohol Survey. Addiction, 89(4), 407-412. doi:10.1111/j.1360-0443.1994.tb00914.x

National Institute on Alcohol Abuse and Alcoholism. What Is Binge Drinking? USA 2021. 

Available from: Binge Drinking | National Institute on Alcohol Abuse and Alcoholism (NIAAA) (nih.gov)

Piano, M. R., Mazzuco, A., Kang, M., & Phillips, S. A. (2017). Cardiovascular Consequences of Binge Drinking: An Integrative Review with Implications for Advocacy, Policy, and Research. Alcohol Clin Exp Res, 41(3), 487-496. doi:10.1111/acer.13329

Rehm, J. (1998). Measuring quantity, frequency, and volume of drinking. Alcohol Clin Exp Res, 22(2 Suppl), 4s-14s. doi:10.1097/00000374-199802001-00002

Rehm, J., Greenfield, T. K., Walsh, G., Xie, X., Robson, L., & Single, E. (1999). Assessment methods for alcohol consumption, prevalence of high risk drinking and harm: a sensitivity analysis. Int J Epidemiol, 28(2), 219-224. doi:10.1093/ije/28.2.219

Stockwell, T., Zhao, J., Chikritzhs, T., & Greenfield, T. K. (2008). What did you drink yesterday? Public health relevance of a recent recall method used in the 2004 Australian National Drug Strategy Household Survey. Addiction, 103(6), 919-928. doi:10.1111/j.1360-0443.2008.02219.x

Reviewers' comments:

Reviewer's Responses to Questions

Comments to the Author

1. Is the manuscript technically sound, and do the data support the conclusions?

Reviewer #1: Yes

Reviewer #2: Yes

2. Has the statistical analysis been performed appropriately and rigorously? 

Reviewer #1: Yes

Reviewer #2: Yes

3. Have the authors made all data underlying the findings in their manuscript fully available?

Reviewer #1: Yes

Reviewer #2: Yes

4. Is the manuscript presented in an intelligible fashion and written in standard English?

Reviewer #1: Yes

Reviewer #2: Yes

5. Review Comments to the Author

Reviewer #1: This is an important systematic review of alcohol consumption and pattern measurement for epidemiological studies of older adults (65 plus years old). The review including 105 studies winnowed from many more in a comprehensive search found wide variation in every aspect of measurement strategy and importantly alcohol use pattern definitions. Basic variance was also found in drink size included, duration of measurement and type of measure, i.e., QF, BSQF, GQF Weekly diary, Last occasion. Conclusions and implications proposed by the authors emphasized the need for standardization. The manuscript is very well written and provides a wealth of tabular material.

I have a number of comments and suggestions for improvement of the paper:

1) LL 104-113 The list of measurement approaches is fine but there is here and in many places following an overuse of secondary references 24 and 36, (WHO international measurement guidelines and SMART, respectively). While I have nothing against these citations based essentially of expert groups, some other citations could help reduce the over-reliance one example being Greenfield & Kerr, Alcohol measurement methodology in epidemiology: recent advances and opportunities- Addiction, 2008. It would be worth emphasizing that last drinking occasion (and yesterday drinking especially) typically have specialized uses for better calibrating longer duration measures such as 12-month measures (e.g., Stockwell et al. What did you drink yesterday? Etc. Addiction, 2008).

Thanks for the valuable suggestion of new articles that could be used. We also appreciate the suggestion about emphasizing the Yesterday method. New text is included which can be read from page 8 and line 186 (Revised Manuscript with Track Changes):

“Especially when linking alcohol consumption with alcohol-related consequences a recall period of at least 12 months is of importance (Dawson & Room, 2000). Shorter recall period is more prone to miss intermittent heavy drinkers (Greenfield & Kerr, 2008). Seasonal variability will also be minimized with 12 months recall period (Greenfield & Kerr, 2008).

The “Yesterday method” is included in the introduction part from page 6 and line 134: 

“….the Yesterday method which asks questions about beverage types and sizes of drinks consumed the day before the interview (Stockwell, Zhao, Chikritzhs, & Greenfield, 2008). 

The advantage of the “Yesterday method” is included from page 7 and line 151:

“The Yesterday method may have some advantages in groups where daily drinking is common (Stockwell et al., 2008). An Australian study of the general population found the Yesterday method to minimize under-reporting of overall alcohol consumption compared to the QF and GQF questionnaires, and recommended the Yesterday method as a supplement to the QF and GQF questionnaires (Stockwell et al., 2008)”. 

In the discussion part the following has been included as now reads from page 73 and line 585: 

“Participants are likely to report drink sizes they actually consume which differ from the size of standard drinks (Greenfield & Kerr, 2008)”. 

2) I commend the authors for identifying LL 59-64 identifying perennial problems of measurement and pattern definition. A type of pattern definition that does not appear to have been given sufficient attention in the excellent review (perhaps because the older age literature has attended to it less than in general population studies), is that of relative binging vs spacing of drinks to obtain the same volume. At the lower weekly levels such as those recommended for older adults by NIAAA 5/4+ (M/W) drinks or (5 or 4 x 12 g = 60 or 40 g ethanol) is a good indicator of this intake variability. At the higher volume levels 8+ may separate spacers (or even drinkers) from bunched (or binge) drinking patterns (per the GQF measure used, however, in only one of the included studies: Satre et al. [53]). Some recommendation for a smaller set of items have noted the value of including maximum in any day (usually categorized).

Thanks for making us aware of the drinking pattern binging versus spacing of drinks. In the introduction part we have added the following which can be read from page 5 and line 95:

“Due to the greater sensitivity to health risk of alcohol among older adults, the prevalence of binge drinking in older age is of interest (Azagba, Shan, Latham, & Manzione, 2020; NIAAA, 2021). NIAAA defines binge drinking as consuming 5 or more drinks among men and 4 or more drinks among women in about two hours (NIAAA, 2021). Assessment of binge drinking is relevant in alcohol surveys of older adults. Furthermore, it may be relevant to distinguish between binging (infrequent heavy) versus spacing (steady daily) drinking patterns (Greenfield, 1998), and especially among older adults drinking higher weekly volume (i.e., 8 drinks or more). These opposite drinking patterns can produce the same weekly alcohol volume (Greenfield, 1998) but binge drinking may lead to higher risk of negative health consequences than steady daily drinking (Liu, Redmond, Morrow, & Cullen, 2011; Piano, Mazzuco, Kang, & Phillips, 2017). In alcohol surveys of older adults, it may also be relevant to ask about the maximum number of drinks consumed in any day, the frequency of subjective drunkenness, drinking context and duration of drinking occasions (Dawson & Room, 2000; Greenfield, 1998; Greenfield & Kerr, 2008)”. 

3) LL 133-137 The lack of standardization – aim of a systematic review somewhat puts the cart before the hose, in the sense that the ultimate finding that there is very little standardization seems to motivate the review that finds this. The conclusions and Implications sections echo the hope for moving toward more standardization, but I think it might be acknowledges somewhere that different aims (as well as traditions and simple research group preferences) result in the variety of measures and definitions seen. The authors could note (relevant although not specific to assessing drinking patterns among older adults) that over the years there have been expert meetings and groups convened to discuss alcohol measurement and patterns, and sometimes to make some consensus recommendations. One meeting organized by Stockwell was in Freemantle AU, and led to reference 24. Another was at NIAAA (See Alcohol Clin Exp Res 22, 4S-14S & 52S-62S, 1998). A third was a conference organized by Room held at Skarpö, Sweden in 2000, resulting in a special edition of the J Substance Use (e.g., Dawson & Room, Toward agreement on ways to measure and report drinking patterns and alcohol-related problems in adult general population surveys: the Skarpö Conference overview. J Subst Abuse 12, 1-21, 2000. SMART [Ref 36] represents a more recent effort. I therefore feel that the conclusions regarding standardization might be tempered by acknowledging the many previous such efforts of groups of alcohol epidemiologists, which have thus far failed to fully achieve such a standardization of alcohol measures and pattern definitions for general population surveys and similarly by extension for the aging subgroup. However, the authors are commended for recognizing that patterns and guidelines could differ in the elder group, and that on guidelines there is no empirically backed international guideline (LL 542-543).

We really appreciate that the reviewer has made us aware of the number of expert groups and meetings that have led to several published studies. These expert groups are cited both in the introduction and discussion part. Please read from page 5 and line 110:

“During the last decades there have been several international expert groups and meetings convened to discuss alcohol measurement and drinking patterns in the general adult general population (Dawson & Room, 2000; Greenfield, 1998; Moskalewicz & Sieroslawski, 2010; Rehm, 1998; WHO, 2000). The aim of these expert groups has been to give an overview of the current knowledge on measuring frequency, quantity, and volume of drinking, and make consensus recommendations.

From page 6 and 138 the following is included: 

The QF questionnaire has been widely used to measure alcohol consumption since the early 1950s (Rehm, 1998). The GQF and the BSQF questionnaires measure both volume of alcohol and patterns of drinking, have been used less, but have an advantage over the QF questionnaire which only measure the volume (Rehm, 1998). Previous studies have reported higher estimates of volume and prevalence of high-risk drinking using GQF compared to QF questionnaire and weekly drinking measures (Midanik, 1994; Rehm et al., 1999). A variation of the QF questionnaire (the ‘period-specific normal week’ assessment instrument) includes questions about drinking variability and asks about alcohol consumption during a normal week the last year (Rehm, 1998). The alcohol consumption during the week is separated between weekdays and on weekend (i.e., Friday, Saturday, and Sunday) (Rehm, 1998). This assessment instrument is relevant to use when exploring groups where weekend drinking may vary substantially from drinking during the week (Rehm, 1998). The Yesterday method may have some advantages in groups where daily drinking is common (Stockwell et al., 2008). An Australian study of the general population found the Yesterday method to minimize under-reporting of overall alcohol consumption compared to QF and GQF questionnaires, and recommended the Yesterday method as a supplement to the QF and GQF questionnaires (Stockwell et al., 2008). 

When it comes to questions about drinking frequency it is preferable to ask in terms of prespecified frequency range categories such as twice a day, daily, 5-6 times a week/nearly every day, 3-4 times a week, 1-2 times a week, 2-3 times a month, once a month, 6-11 times a year, and 1-5 times a year (Dawson & Room, 2000). Furthermore, it is recommended to ask the question in terms of number of drinks per day and not per occasion, since a day may be a more ‘objective’ measure (Dawson & Room, 2000). Continued drinking past midnight should be defined in the day (Dawson & Room, 2000).

The introduction to the aim of the study is changed and can be read from page 8 and line 192: 

“Even though there have been several previous efforts regarding the standardization of methods to assess, define, and measure alcohol consumption in the adult general population (Dawson & Room, 2000; Greenfield, 1998; Moskalewicz & Sieroslawski, 2010; Rehm, 1998; WHO, 2000), the standardization has so far almost been absent for the aged population. It is important to increase the attention around the need for standardized methodology in alcohol surveys in the aged population. Thus, the aim of this study is to systematically review methods used in epidemiological studies to define drinking patterns and measure alcohol consumption among older adults”.

Some new elements regarding drinking context are included in the discussion section from page 74 and line 607:

“In alcohol surveys, it is recommended to ask about the drinking context, which focuses on drinking with meals or not, drinking alone or not (e.g., with family members, friends, work colleagues, etc.), drinking on a weekday or on a weekend, and drinking in public (bars and restaurants) or at home (Dawson & Room, 2000; Moskalewicz & Sieroslawski, 2010; WHO, 2000).

We acknowledge the expert groups under “Implications” in the discussion part (see page 75 and from line 616):

“We want to acknowledge the previous expert groups and alcohol epidemiologists for their effort to standardize the alcohol methodology in adult general population surveys (Dawson & Room, 2000; Greenfield, 1998; Moskalewicz & Sieroslawski, 2010; Rehm, 1998; WHO, 2000). However, it seems thus far that they have failed to fully achieve a standardization, and especially for the subgroup of the aged population. Different aims, traditions, and simple research group preferences may have resulted in the variety of measures and definitions found in this systematic review… Methodological studies are needed to study the reliability and validity of different assessment instrument and definitions (Rehm et al., 1999).

In the conclusion we also refer to the expert groups and the alcohol epidemiologists (see page 75 and line 632):

“Several previous expert groups and alcohol epidemiologists have had an aim of standardizing the alcohol methodology in adult general population surveys. However, so far it seems that they have failed to fully achieve a standardization, and especially in the subgroup of the aged population….Different aims, traditions, and simple research group preferences may have resulted in the variety of measures and definitions found in this systematic review”:

4) LL 483-484 An important reference that complements [134] is Rehm, et al. Assessment methods for alcohol consumption, prevalence of high risk drinking and harm: a sensitivity analysis. Int J Epidemiol 28, 219-224, 1999. Even though GQF was only used in one paper in the review, this proved superior to QF and weekly diary in capturing risky and harmful drinking volumes.

Thanks for the suggestion of this important reference. In the discussion the following is included (see page 69 and line 492): 

“A previous study by Rehm et al. (Rehm et al., 1999) has also shown that the GQF questionnaire was superior to QF questionnaire and weekly diary in capturing risky and harmful drinking volumes”.

Reviewer #2: Dr. Tevik and collaborators reported in an elegant systematic review how different are the parameters used for the categorization/assessment of alcohol drinking among epidemiological studies. This paper raised an important question since the assessment of alcohol consumption is used for the diagnostic of alcohol use disorder, thus if studies/hospitals don’t use the same criteria and definitions the diagnostics could be under or overestimated. All the criteria used for the studies selection/inclusion were well thinking and organized. Authors also followed a recommended structure for systematic review (prism9). Data is well reported and clear in all tables throughout the manuscript. All important “side” points were mentioned, such as problems with stigma; underestimation of alcohol consumption during self-report; the population’s knowledge about the definitions of standard-drinking and the use of figures to clarify that during interviews. The limitations and strengths were also mentioned. My two suggestions are: 

1) Flip the title: A systematic review of self-report measures used in epidemiological studies to assess alcohol consumption among older adults; 

Thanks for the suggestion. We have changed the title (see page 1, line 1). 

2) Although authors included the country of each study on table 3, would be also interesting if they keep that information on table 6, so the readers would have an idea about which are the criteria used in each country. 

We agree, and the country is included for each study in Table 3 (page 43). 

Besides that, the study is timely, the methodology applied is sound, the authors provide clear models and hypothesis, and the findings are clearly embedded in a theoretical framework in the discussion section.

6. PLOS authors have the option to publish the peer review history of their article (what does this mean?). If published, this will include your full peer review and any attached files.

Do you want your identity to be public for this peer review? For information about this choice, including consent withdrawal, please see our Privacy Policy.

Reviewer #1: No

Reviewer #2: No

We have uploaded Figure 1 to PACE to ensure that the figure met PLOS requirements.

References

Azagba, S., Shan, L., Latham, K., & Manzione, L. (2020). Trends in Binge and Heavy Drinking among Adults in the United States, 2011-2017. Subst Use Misuse, 55(6), 990-997. doi:10.1080/10826084.2020.1717538

Dawson, D. A., & Room, R. (2000). Towards agreement on ways to measure and report drinking patterns and alcohol-related problems in adult general population surveys: the Skarpö conference overview. J Subst Abuse, 12(1-2), 1-21. doi:10.1016/s0899-3289(00)00037-7

Greenfield, T. K. (1998). Evaluating competing models of alcohol-related harm. Alcohol Clin Exp Res, 22(2 Suppl), 52s-62s. doi:10.1097/00000374-199802001-00008

Greenfield, T. K., & Kerr, W. C. (2008). Alcohol measurement methodology in epidemiology: recent advances and opportunities. Addiction, 103(7), 1082-1099. doi:10.1111/j.1360-0443.2008.02197.x

Liu, W., Redmond, E. M., Morrow, D., & Cullen, J. P. (2011). Differential effects of daily-moderate versus weekend-binge alcohol consumption on atherosclerotic plaque development in mice. Atherosclerosis, 219(2), 448-454. doi:10.1016/j.atherosclerosis.2011.08.034

Midanik, L. T. (1994). Comparing usual quantity/frequency and graduated frequency scales to assess yearly alcohol consumption: results from the 1990 US National Alcohol Survey. Addiction, 89(4), 407-412. doi:10.1111/j.1360-0443.1994.tb00914.x

Moskalewicz J, Sieroslawski J. Drinking poulation surveys – Guidance document for 

standardized approach. Final report prepared for the project Standardizing Measurement of Alcohol-Related-Troubles – SMART. Institute of Psychiatry and Neurology, Warsaw 2010. Available from: https://www.drugsandalcohol.ie/15682/1/EU_Comm_Drinking_population_surveys.pdf

National Institute on Alcohol Abuse and Alcoholism. What Is Binge Drinking? USA 2021. 

Available from: Binge Drinking | National Institute on Alcohol Abuse and Alcoholism (NIAAA) (nih.gov)

Piano, M. R., Mazzuco, A., Kang, M., & Phillips, S. A. (2017). Cardiovascular Consequences of Binge Drinking: An Integrative Review with Implications for Advocacy, Policy, and Research. Alcohol Clin Exp Res, 41(3), 487-496. doi:10.1111/acer.13329

Rehm, J. (1998). Measuring quantity, frequency, and volume of drinking. Alcohol Clin Exp Res, 22(2 Suppl), 4s-14s. doi:10.1097/00000374-199802001-00002

Rehm, J., Greenfield, T. K., Walsh, G., Xie, X., Robson, L., & Single, E. (1999). Assessment methods for alcohol consumption, prevalence of high risk drinking and harm: a sensitivity analysis. Int J Epidemiol, 28(2), 219-224. doi:10.1093/ije/28.2.219

Stockwell, T., Zhao, J., Chikritzhs, T., & Greenfield, T. K. (2008). What did you drink yesterday? Public health relevance of a recent recall method used in the 2004 Australian National Drug Strategy Household Survey. Addiction, 103(6), 919-928. doi:10.1111/j.1360-0443.2008.02219.x

World Health Organization. International guide for monitoring alcohol consumption and 

related harm. Department of Mental Health and Substance Dependence,

Noncommunicable Diseases and Mental Health Cluster, World Health Organization, 2000. WHO/MSD/MSB/00.4. Available from: https://apps.who.int/iris/handle/10665/66529

---

## [Decision Letter · Decision Letter 1]

29 Nov 2021

A systematic review of self-report measures used in epidemiological studies to assess alcohol consumption among older adults

PONE-D-21-29079R1

Dear Dr. Tevik,

We’re pleased to inform you that your manuscript has been judged scientifically suitable for publication and will be formally accepted for publication once it meets all outstanding technical requirements.

Kind regards,

Petri Böckerman

Academic Editor

PLOS ONE

Additional Editor Comments (optional):

Reviewers' comments:

Reviewer's Responses to Questions

**Comments to the Author**

1. If the authors have adequately addressed your comments raised in a previous round of review and you feel that this manuscript is now acceptable for publication, you may indicate that here to bypass the “Comments to the Author” section, enter your conflict of interest statement in the “Confidential to Editor” section, and submit your "Accept" recommendation.

Reviewer #1: All comments have been addressed

Reviewer #2: All comments have been addressed

2. Is the manuscript technically sound, and do the data support the conclusions?

Reviewer #1: Yes

Reviewer #2: Yes

3. Has the statistical analysis been performed appropriately and rigorously? 

Reviewer #1: Yes

Reviewer #2: Yes

4. Have the authors made all data underlying the findings in their manuscript fully available?

Reviewer #1: Yes

Reviewer #2: Yes

5. Is the manuscript presented in an intelligible fashion and written in standard English?

Reviewer #1: Yes

Reviewer #2: Yes

6. Review Comments to the Author

Reviewer #1: Comments form reviewers were thoroughly addressed in the R1 submission and I am satisfied that the MS is now in excellent shape. Thank you for taking such good account of the feedback.

Reviewer #2: All of my comments were addressed by the authors, and I think the paper is suitable for publication.

7. PLOS authors have the option to publish the peer review history of their article (what does this mean?). If published, this will include your full peer review and any attached files.

Reviewer #1: No

Reviewer #2: No

---

## [Editor Report · Acceptance letter]

3 Dec 2021

PONE-D-21-29079R1 

A systematic review of self-report measures used in epidemiological studies to assess alcohol consumption among older adults 

Dear Dr. Tevik:

I'm pleased to inform you that your manuscript has been deemed suitable for publication in PLOS ONE. Congratulations! Your manuscript is now with our production department. 

Kind regards, 

on behalf of

Professor Petri Böckerman 

Academic Editor

PLOS ONE